# Harmonizing Multi-Source Sonar Backscatter Datasets for Seabed Mapping Using Bulk Shift Approaches

**Benjamin Misiuk [1,*] , Craig J. Brown [2], Katleen Robert [1] and Myriam Lacharité [3]**

1   School of Ocean Technology, Fisheries and Marine Institute of Memorial University of Newfoundland, St. John's, NL A1C 5R3, Canada; katleen.robert@mi.mun.ca

2   Department of Oceanography, Dalhousie University, Halifax, NS B5H 4R2, Canada; craig.brown@dal.ca

3   Institute for Marine and Antarctic Studies, University of Tasmania, Hobart 7053, Australia; myriam.lacharite@utas.edu.au

*   Correspondence: bmisiuk@mun.ca; Tel.: +1-709-778-0584

**Abstract:** The development of multibeam echosounders (MBES) as a seabed mapping tool has resulted in the widespread uptake of backscatter intensity as an indicator of seabed substrate properties. Though increasingly common, the lack of standard calibration and the characteristics of individual sonars generally produce backscatter measurements that are relative to a given survey, presenting major challenges for seabed mapping in areas that comprise multiple MBES surveys. Here, we explore methods for backscatter dataset harmonization that leverage areas of mutual overlap between surveys for relative statistical calibration—referred to as "bulk shift" approaches. We use three multispectral MBES datasets to simulate the harmonization of backscatter collected over multiple years, and using multiple operating frequencies. Results suggest that relatively simple statistical models are adequate for bulk shift harmonization procedures, and that more flexible approaches may produce inconsistent results that risk statistical overfitting. While harmonizing datasets collected using the same operating frequency from separate surveys is generally feasible given reasonable temporal limitations, results suggest that the success at harmonizing datasets of different operating frequencies partly depends on the extent to which the frequencies differ. We recommend approaches and diagnostics for ensuring the quality of harmonized backscatter mosaics, and provide an R function for implementing the methods presented here.

**Keywords:** backscatter; multispectral; multibeam; echosounder; seabed mapping; benthic; habitat mapping

---

## 1. Introduction

Multibeam echosounders (MBES) have recently become the preferred survey tool for seabed mapping. The high density of soundings, which are deployed in a swath pattern that is typically several times wider than the water depth, makes them ideal for efficient hydrographic charting [1,2]. Additionally, MBES record measurements on acoustic reflectivity, referred to as "backscatter", which can be used to derive information regarding the nature of the seafloor [3], such as volume heterogeneity (e.g., sediment grain size, distribution, and geological layering) and interface characteristics (e.g., substrate, roughness, bedforms; [4,5]). Therefore, the MBES signal provides both qualitative and quantitative information on seafloor environmental characteristics, and it has been commonly used to delineate and map surficial geology (e.g., [6–9]). As an extension of this application, relationships between benthic species and physical seafloor characteristics have allowed backscatter to be utilized as a tool for describing benthic habitat. A growing number of studies are now using backscatter, often

combined with seabed morphology information derived from bathymetric measurements, to study and map benthic species distributions and biodiversity patterns on the seafloor [10–12].

The interpretation of backscatter intensity is complex when compared to bathymetric measurements [5], which are relatively straightforward to derive using the geometry and transit time of the acoustic signal [1]. This is commonly recognized as a limitation of using backscatter as a proxy for seafloor geology or benthic habitat [3,10,12]. There are several confounding factors that need to be considered for backscatter intensity to represent seafloor material characteristics:

1.  The MBES electronics that are involved in transmitting and receiving the acoustic signal, and how they measure acoustic intensity [13]. This commonly varies between sonar manufacturers, between successive generations of sonars from the same manufacturer, and even between different units of the same sonar from the same manufacturer. Information on how these measurements are derived is often proprietary and unknown by the end user of the data [5].
2.  Propagation loss of the acoustic signal in the water column. This is affected by a complex interaction between temporally and geographically dynamic oceanographic parameters, such as temperature and density, and the presence of water column scatterers, such as particulate material, plankton, and nekton, which are often not fully quantified during data acquisition [13,14].
3.  Frequency-dependent interaction between the acoustic signal and substrate. The depth of substrate penetration and the reflection and scattering of the signal at the seabed depend on sediment characteristics and co-depend on grazing angle (see point 4 below), but will also change with operating frequency [13,15].
4.  The angular dependence of the backscatter response, which co-depends on substrate material characteristics. For example, a hard/rough seabed with coarse sediments scatters the acoustic signal in all directions, yielding a backscatter intensity that is largely independent of the grazing angle of individual beams across the MBES swath. However, a soft/smooth surface comprising fine sediments produces less scattering, resulting in a maximum backscatter intensity at nadir and progressively lower returns with increasing grazing angle of the outer beams [13,16].

Geometric and radiometric corrections that account for the factors that are outlined above can now be readily applied via commercial MBES post-processing software, allowing for the production of compensated backscatter mosaics for geological interpretation [17]. Alternatively, some studies have sought to preserve the angular dependence of the backscatter response to inform seafloor classification [16,18,19]. Regardless of how these challenges are handled, the goal is generally to produce backscatter values that consistently represent measurable seafloor characteristics. A singular unified solution is difficult to achieve though, owing to the large number of confounding factors that affect the calculation of backscatter intensity.

Obtaining absolute measurements of seafloor backscatter is a particularly challenging task. Ideally, backscatter measurements should be calibrated to combine or compare both spatial and temporal datasets from different surveys [5]. There are two types of MBES backscatter calibrations: absolute and relative. Absolute calibration should preferably be handled by the sonar manufacturer under controlled conditions, where transducer and electronic components are measured to determine frequency response, angular directivity, level linearity, and other relevant parameters for both signal transmission and reception. In practice, this is often not adequately undertaken and no quality standards are currently available for backscatter data—variability is commonly observed between MBES systems [5,20]. Relative calibration has become a practical option in the absence of absolute calibration, and a number of methods have recently been explored including the use of natural reference areas [20,21], calibration targets [22], and comparison against calibrated single-beam sonar backscatter collected simultaneously [23,24]. At present, there are no standard, widely accepted approaches to calibration. The large volume of non-calibrated legacy data will remain relative even if calibration standards are eventually adopted, posing challenges for the use of such datasets in benthic habitat and surficial geology mapping studies.

Seafloor mapping can involve combinations of separate surveys that utilize multiple sonar systems, potentially of various operating frequencies, sometimes over multiple years (e.g., [25,26]). Because standard calibration procedures have not been widely adopted, and owing to the inherent complexities of measuring backscatter return, simply combining backscatter data from separate surveys or systems for use in seabed mapping is likely to introduce large amounts of error when using automated approaches, which can ultimately impact seabed classifications or the quality of statistical models. Furthermore, multi-source backscatter datasets are becoming increasingly common with the maturation of MBES as a technology [12], and they are an invaluable resource given the high cost of acquisition. Therefore, it is necessary to develop robust methods for using multi-source backscatter for geological and biological seabed mapping, but few studies have addressed this challenge. Notable exceptions have generally employed strategies to avoid the integration and harmonization of different backscatter datasets to respect frequency-dependent differences in the sediment response for MBES [26] and side-scan [27] datasets. Such approaches are only feasible with a low number of survey datasets, or with datasets of similar extent, combined with adequate ground-truthing within each survey coverage.

Combining multiple backscatter datasets to generate a single harmonized mosaic is a potential approach for dealing with multi-source backscatter data. The overlap between surveys provides a common area at which to compare backscatter intensities and derive statistical relationships between datasets. Predictive surficial geology and habitat modelling generally rely on continuous-coverage environmental data layers to produce map products, and a single harmonized backscatter mosaic is therefore often desirable (e.g., [9,28]). An uneven distribution of ground truth over separate backscatter datasets inhibits the production of full-coverage habitat maps and enhances uncertainty in the map products [26]. Harmonizing backscatter datasets to produce a single mosaic prior to habitat mapping would ameliorate some of these difficulties. Methods to facilitate the combination of several partially overlapping backscatter datasets have been applied by Hughes Clarke et al. [25] in the Bay of Fundy, Canada, wherein offsets were applied to the values of one backscatter layer to match another. Such "bulk shift" approaches to multi-source backscatter harmonization have not otherwise been widely developed.

Here, we explore methods for harmonizing disparate backscatter datasets for use in seabed mapping. The recent implementation of multispectral MBES—collecting data at multiple frequencies simultaneously to produce multiple backscatter datasets [15,29]—provides new opportunities to simulate the application of multi-source harmonization methods. In this study, we use multispectral MBES datasets to simulate the combination of backscatter data from separate surveys, wherein the error of harmonized datasets can be measured against actual values that were obtained from different years or from different operating frequencies. The goals of this paper are to:

1. assess the feasibility of harmonizing backscatter datasets from different surveys obtained using the same MBES system and operating frequency, and also using different operating frequencies, using bulk shift methodologies at varying levels of complexity;
2. compare several bulk shift methods for harmonizing backscatter datasets collected from different surveys using the same operating frequency, and with different operating frequencies; and,
3. provide general recommendations for harmonizing multi-source backscatter datasets, with an R function for implementing these methods and evaluating the results based on the findings in this paper.

## 2. Materials and Methods

Three multispectral MBES datasets were used in this study to investigate the harmonization of relative backscatter obtained using three different operating frequencies for two different sites, and over two years: (1) Bedford Basin collected in 2016, (2) Bedford Basin in 2017, and (3) Patricia Bay collected in 2017 (Figure 1). All three datasets were collected using an R2Sonic 2026 MBES (two different MBES units were used in 2016 and 2017), operating at 100, 200, and 400 kHz frequencies simultaneously. The MBES was integrated with a Valeport sound velocity probe mounted adjacent to the sonar head, and

POS MVWave Master Inertial Navigation System (INS), utilizing two Trimble GPS antennas. All of the systems were integrated through QPS QINSy software installed on the acquisition PC aboard the wheelhouse of the survey vessel. CTD casts at the time of survey were conducted using an AML Base X2 that was fitted with a set of conductivity, temperature, and pressure probes. Data were processed using the QPS software suite with a consistent workflow for each site.

### 2.1. Patricia Bay, British Columbia

The physiography and surficial geology of Patricia Bay has been described previously [30]. The multispectral MBES survey covers a site in the central section of the bay (Figure 1a), ranging in depth from 20 to 72 m, deepening from northeast to southwest. The seabed along the centre of the survey, following the long axis, is expected to be muddy, flanked on both sides by patches of sand then gravel at the survey margins [30]. These patches are visible in each of the multispectral frequencies, but there seem to be greater differences in relative backscatter between sediment types at the lowest frequency (100 kHz) than the highest (400 kHz), which appears to be more homogenous.

### 2.2. Bedford Basin, Nova Scotia

The two Bedford Basin datasets cover approximately the same area of seafloor at the mouth of the basin (Figure 1b), where depths range from 14 to 63 m. The physiography has been described extensively [15,31]. A shallow ridge at around 15 m depth in the southern part of the survey contains coarse, hard sediment (bedrock and gravel, with boulder-sized clasts). Depth increases to the north, where the sediments are softer and the seafloor becomes relatively flat. Visible differences between backscatter frequencies in the deeper part of the survey are believed to be a result of subsurface dredge spoil, smothered by fine sediments [15].

### 2.3. MBES Data Processing

Bathymetry and backscatter data quality were monitored during acquisition, and post-processing was conducted using the QPS software suite. The Fledermaus Geocoder Toolbox (FMGT) was used to process multispectral backscatter data. Absorption coefficients were calculated to correct for water column attenuation of each frequency using CTD casts at each area, and all frequency-specific corrections (e.g., pertaining to beam widths, etc.) were applied automatically in FMGT. By extracting frequency-specific pings from the R2Sonic 2026 data in FMGT, separate corrected backscatter mosaics were generated for 100, 200, and 400 kHz frequencies, which represent the relative backscatter intensity without the angular dependence of beam incidence on the seafloor. Bathymetric data were corrected for tides and manually cleaned in QPS Qimera. Each data layer was exported as a 0.5 m ASCII grid file.

### 2.4. Simulating Harmonization

Each study area was divided into two parts that overlapped by approximately one full survey line to simulate the combination of datasets that were acquired during separate surveys. For each study site, one of the two sections were treated as the reference area, containing the "target" backscatter dataset, while the other section was treated as the test area, containing the "shift" backscatter dataset (Figure 1). The overlap between the target and shift datasets represents the information that is normally available to the mapper for harmonizing datasets from different surveys. The true accuracy of the correction can be determined by comparing the corrected shift layer to the portion of the target dataset that was not used to inform the correction, since each frequency of the multispectral datasets has the same extent (i.e., the target data within the test area).

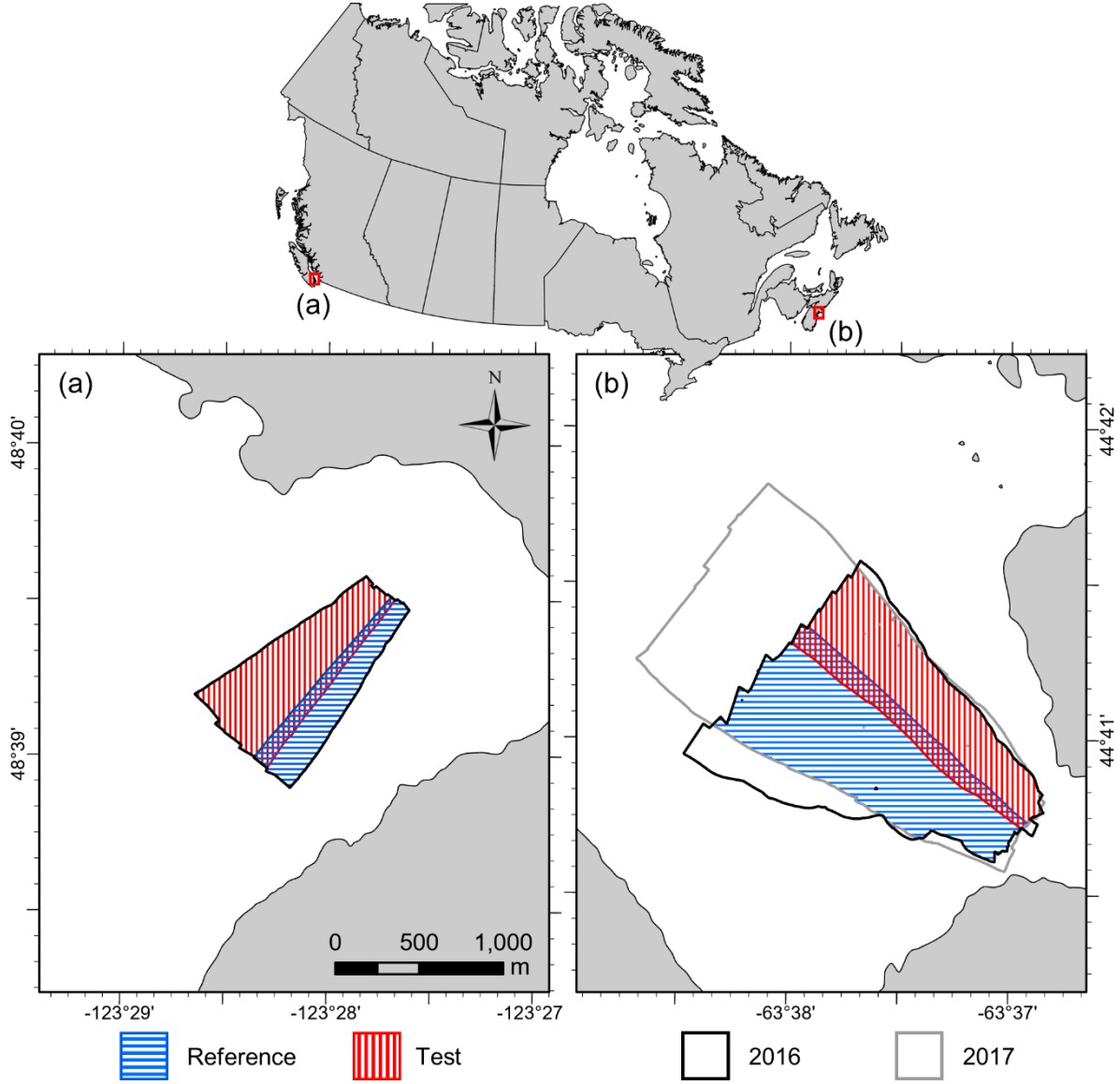

**Figure 1.** Locations of (**a**) Patricia Bay, British Columbia, and (**b**) Bedford Basin, Nova Scotia, Canada. Blue and red hatches delimit reference and test areas used for bulk shift simulations at Patricia Bay in 2016, and Bedford Basin in 2016 and 2017. Test areas contain the "shift" datasets, which are compared to the withheld portion of the "target".

Following Hughes Clarke et al. [25], we will refer broadly to the correction of entire backscatter raster datasets as "bulk shift" approaches. If several key assumptions can be accepted, it may be feasible to use the areas where MBES surveys overlap to calibrate their relative backscatter measurements, which can be used to standardize the datasets to produce a single harmonized output. First, if the goal is to generate an internally consistent relative proxy for seabed substrate, then we must accept that the backscatter measurements of each dataset are a function of the same substrate properties. We must also accept that the values obtained within each survey are internally consistent (i.e., that the same substrate produces the same backscatter response). Finally, there must be an acceptable level of temporal homogeneity between separate surveys—large amounts of natural variability throughout time will preclude any attempts at harmonization. If these assumptions are tenable, attempts can be made to adjust the relative values of one backscatter dataset to match the scale of another.

Here, the error (i.e., the difference) between the target and shift datasets from the area of overlap was treated as the response variable when testing bulk shift methods, which was preferred to treating the shift dataset as the response for several reasons. First, it facilitates the visualization of the location and magnitude of error as a function of the dataset being shifted (e.g., Figure 2), which may help considerably in conceptualizing the relationships between the datasets being combined, and therefore in selecting an appropriate bulk shift method. This can also inform on the effects that the shift will have on specific data values, which is relevant, for instance, when the full dynamic range of the shift dataset is not represented in the area of overlap. How a given bulk shift method handles extrapolation of the error at these values can have important consequences for the final harmonized mosaic. Second, the primary map product of bulk shifts that use the error between backscatter layers as the response will be the map of predicted error (i.e., the correction values) for the shift layer, which makes explicit the locations where corrections are occurring if the bulk shift method is non-uniform. This can be useful for identifying the effects that the shift had on the detection of specific features or substrate types, and for assessing limitations of the correction.

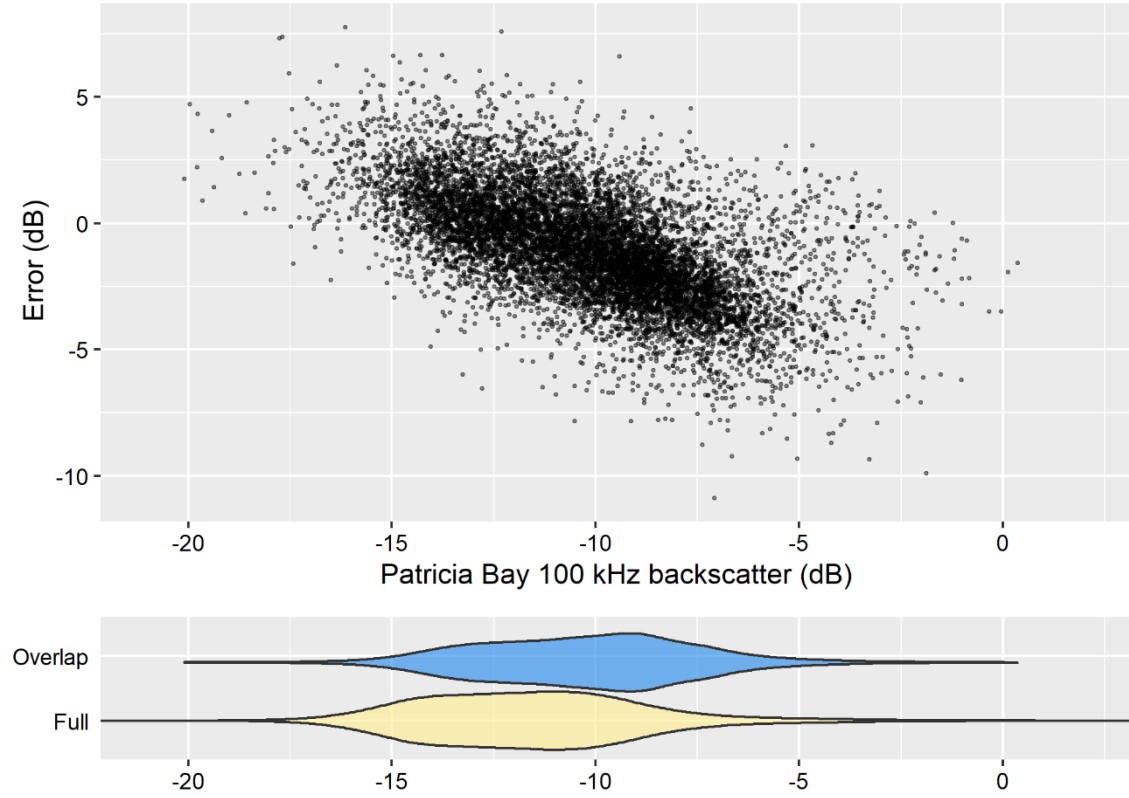

**Figure 2.** Error between 100 and 200 kHz Patricia Bay backscatter datasets from the area of overlap as a function of the 100 kHz signal. Violin plots show the distribution of the 100 kHz data from the area of overlap (blue), which corresponds to the error plot, compared to the full distribution of the full 100 kHz dataset (yellow).

A consistent workflow was followed for each bulk shift simulation consisting of five steps. Steps 1–4 can be performed outside of simulation; step 5 is the simulation evaluation:

1.  Visualization: backscatter values where the datasets overlap are plotted, and the error between datasets is plotted to observe and diagnose obvious relationships (e.g., Figure 2). Filters such as locally estimated scatterplot smoothing (LOESS) can be used as a visual aid.
2.  Model fitting: a model is fit to the error between datasets. The fit is assessed statistically and visually against plots from 1 above.

3.  Shifting: the fitted model is used to predict the error across the shift dataset. The prediction is mapped, visually assessed, and added to the shift dataset to obtain a corrected layer with respect to the target.
4.  Mosaicking: the corrected shift layer is mosaicked with the target to produce a single harmonized mosaic.
5.  Evaluation: the corrected shift layer values are compared to the portion of the target dataset withheld for testing to produce the test statistics. The test error is also mapped and visually assessed to aid in determining the quality and limitations of the bulk shift. The harmonized mosaic is visually compared to the full original target dataset.

An R function is included (Supplementary Code S1) that automates the harmonization of two backscatter raster datasets using this process (steps 1–4), also returning the diagnostics that are presented in this study. The function allows for the use of all modelling methods presented here, facilitates alternative modelling approaches, and includes options for tuning model parameters. Additionally, it supports any number of covariates, subsampling for large datasets, and optionally returns diagnostic two-dimensional (2D) and interactive three-dimensional (3D) plots. Details and guidelines for harmonizing backscatter datasets using this function are provided in Supplementary Document S1 as a tutorial.

Modelling methods spanning a range of complexities were tested for each simulated bulk shift (step 2 above). The original 0.5 m backscatter grids were resampled to 1 m resolution (using a mean aggregation) to reduce the sizes of raster files. The area of overlap between the datasets still contained ~180,000 and 80,000 cells for the Bedford Basin and Patricia Bay datasets, respectively. 10,000 cells from each overlap were randomly sampled for statistical modelling to facilitate computational speed. At the simplest, the mean of the error between the two layers was added to the shift layer (i.e., an intercept-only model), which assumes that the difference between backscatter responses is constant across the dynamic range of the dataset. Simple and multiple linear regression (SLR and MLR, respectively) were used to test for non-uniform error across the range of values being corrected in the shift dataset. Other regression approaches that can accommodate non-linearity or polynomials might also be used to model non-linear error (see Appendix A), but these may require tuning for each application. Boosted regression trees (BRTs) were used for their flexibility in modelling non-linear errors, and hyperparameters were held constant throughout all tests.

Exploratory analysis suggested that, although error between backscatter datasets was sometimes monotonically related to the dB level of the shift dataset (e.g., Figure 2), this was not always the case (e.g., Figure 3a). This suggested that error might not always be predicted by backscatter alone, and that other variables might influence the discrepancy between datasets. For example, the error between 400 and 200 kHz datasets from Patricia Bay was also linearly related to bathymetry (Figure 3b), and this relationship appeared to be stronger than that of the backscatter datasets alone (Figure 3c).

Therefore, bathymetry was also tested to explain error between backscatter datasets. It was implemented as an independent predictor in SLR, as a covariate in MLR, and as a covariate in BRTs. To test and control for potential overfitting, the addition of bathymetry to BRT models was also tested without interaction by first modelling the relationship between bathymetry and error using SLR, and then modelling the residual error using BRTs to produce an additive model. Other morphometric variables expected to influence backscatter response might also be included (e.g., roughness, slope; but see [32]), depending on the study area, MBES system, and application. Table 1 summarizes the methods tested here.

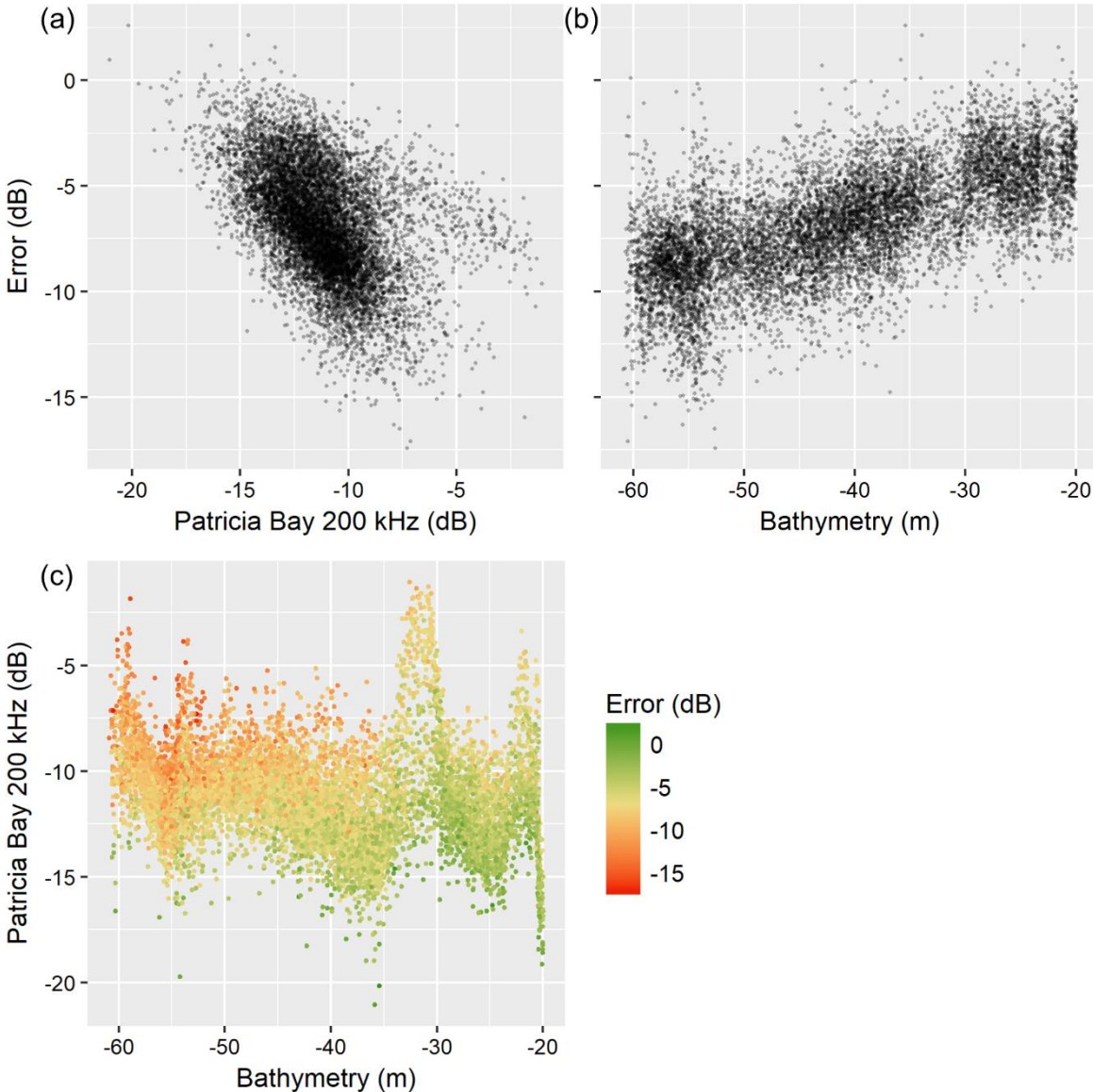

**Figure 3.** Error between 400 and 200 kHz Patricia Bay datasets as a function of (**a**) the 200 kHz backscatter, (**b**) bathymetry, and (**c**) both.

**Table 1.** Bulk shift methods tested.

| Method | Predictors | Interaction | Abbreviation |
| --- | --- | --- | --- |
| Mean | Backscatter | - | - |
| Simple linear regression (SLR) | Backscatter | - | SLR(back) |
| Simple linear regression (SLR) | Bathymetry | - | SLR(bath) |
| Multiple linear regression (MLR) | Backscatter, bathymetry | No | MLR |
| Boosted regression trees (BRT) | Backscatter | - | BRT(back) |
| Boosted regression trees (BRT) | Backscatter, bathymetry | No | BRT(back+bath) |
| Boosted regression trees (BRT) | Backscatter, bathymetry | Yes | BRT(back*bath) |

Backscatter values from each operating frequency differed between the 2016 and 2017 Bedford Basin datasets (Figure 4a), which were collected using two separate R2Sonic 2026 units. The first simulations attempted to harmonize 100, 200, and 400 kHz datasets that were acquired by surveys from separate years in the Bedford Basin using each of the methods in Table 1. Similarly, data that were collected during the same survey, but using different frequencies, also differed (Figure 4b).

The harmonization of datasets of different frequency from the same year was simulated for every combination of datasets for both the Bedford Basin and Patricia Bay datasets using each method in Table 1. Table 2 summarizes the bulk shift simulations.

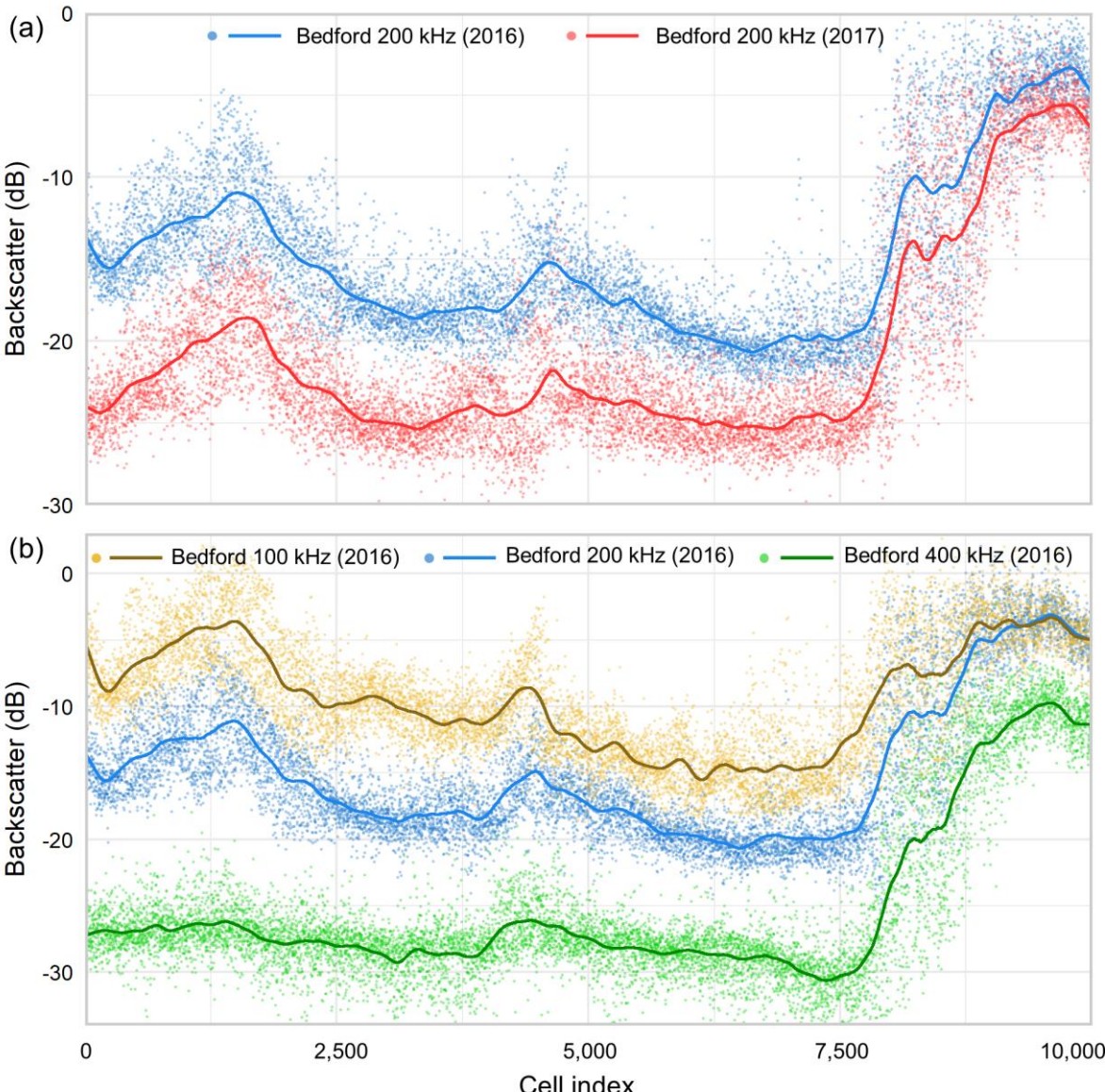

**Figure 4.** (**a**) 200 kHz backscatter data from Bedford Basin in 2016 and 2017 from simulated area of overlap. (**b**) Multispectral backscatter from Bedford Basin in 2016 from simulated area of overlap. X-axes are ordered by cell index, in which the rasters are read; roughly north to south. Dark lines are first order polynomial LOESS curves fit with span of 0.03.

**Table 2.** Summary of bulk shift simulations.

| Target Dataset | Shift Dataset | Location |
| --- | --- | --- |
| 100 kHz 2016 | 100 kHz 2017 | Bedford Basin |
| | 200 kHz 2016 | Bedford Basin, Patricia Bay |
| | 400 kHz 2016 | Bedford Basin, Patricia Bay |
| 200 kHz 2016 | 100 kHz 2016 | Bedford Basin, Patricia Bay |
| | 200 kHz 2017 | Bedford Basin |
| | 400 kHz 2016 | Bedford Basin, Patricia Bay |
| 400 kHz 2016 | 100 kHz 2016 | Bedford Basin, Patricia Bay |
| | 200 kHz 2016 | Bedford Basin, Patricia Bay |
| | 400 kHz 2017 | Bedford Basin |

*2.5. Analysis and Comparison*

Statistical and visual analyses were conducted for each bulk shift simulation. The first goal was to assess the feasibility of harmonizing backscatter datasets from different surveys that were obtained using the same MBES system and operating frequency, and to assess the feasibility of harmonizing backscatter datasets that were obtained using different operating frequencies. The second was to compare several methods for harmonizing these datasets to determine whether some are consistently robust, and are generally preferable.

Two statistics were used to compare the relative quality of the bulk shift methods. The mean absolute error (MAE) between the corrected shift layer and the target layer at the test area (including the overlap; Figure 1) was calculated to measure the average error of the bulk shift over all of the raster cells. This describes the error between the shifted and target layers after correction but provides no information on the distributions of the data, which has implications for the quality of the final mosaic. The two-sample Kolmogorov–Smirnov (K-S) statistic, $D$, measures the difference between cumulative distribution functions (CDF) for continuous variables (e.g., Figure 5). $D$ was calculated to estimate how closely distributions of the shifted datasets matched the target, where smaller values indicate greater similarity.

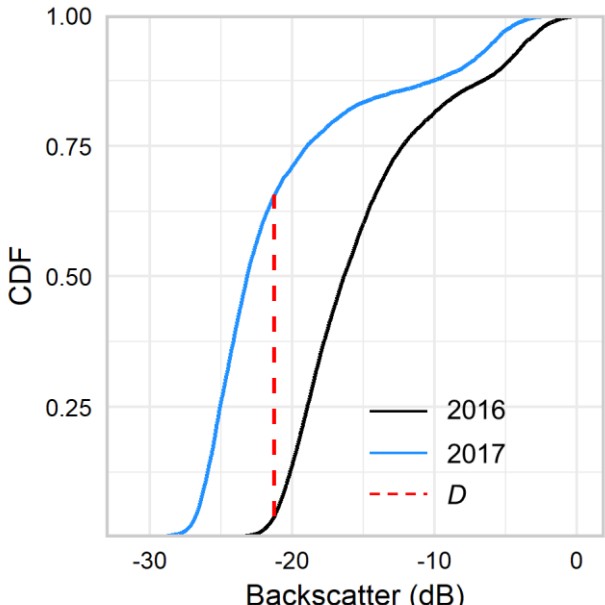

**Figure 5.** Cumulative distribution functions for Bedford Basin 200 kHz datasets at the test area, with $D = 0.62$.

In practice, the quality of a bulk shift will often be evaluated on the "fitted" data, where surveys overlap, yet it is important that those statistics also be representative of the full dataset, outside the area of overlap. Fitting a model that is too specific can cause "overfitting", producing over-optimistic evaluation statistics when compared to its actual performance, and hindering its transferability to new data—in this case, the full backscatter dataset being corrected. The differences between the MAE and *D* calculated on the withheld test data and those calculated on the fitted data at the area of overlap were quantified to determine whether the fitted statistics of a given bulk shift method are representative of its actual performance, or if the method tends to overfit. These differences are denoted $\theta$, wherein a positive value indicates overfitting (the fitted statistics seem better than the test), and an equal or negative value indicates accurate or conservative estimates of performance. Ideally, $\theta$ should be negative, indicating that a given model and its fitted evaluation statistics are transferable to the full dataset.

Each bulk shift method (Table 1) was ranked according to the statistical criteria to facilitate their comparison. The ranks for MAE and *D* were assigned based on the average rank relative to the other methods across all simulations for (i) harmonization of the Bedford Basin datasets between years with the same operating frequency, and (ii) the harmonization of Bedford Basin and Patricia Bay datasets of different operating frequencies. The differences between evaluation statistics of the fitted and test data ($\theta$) were used to assign ranks for overfitting, describing how well the fitted statistics represented the quality of the bulk shifts as compared to the target dataset in test area. If, after correction, the value of the test statistic was less than or equal to that of the fitted, then the estimate was considered to be conservative, and it was assigned the best rank (tied with any other method demonstrating conservative estimates). The ranks for $\theta$ of each statistic were averaged across all simulations for comparison.

The quality of each bulk shift harmonization was also visually assessed. The fit of each model to the backscatter error was visualized using 2D plots for bivariate models (i.e., only backscatter or bathymetry) and 3D plots for multivariate models to assist in understanding how the model corrections affect the data being shifted. The distribution of test error for the shifted layer was mapped by comparing it to the withheld portion of the target layer, and the final harmonized mosaic was visually compared to the original target layer (see Appendix C).

## 3. Results

### 3.1. Harmonizing Different Surveys (Same Frequency)

MLR and BRT(back+bath) produced corrected shift layers with the lowest MAE, followed by SLR(bath) (Table 3). The lowest MAEs achieved using these methods were 1.6, 1.6, and 1.9 dB for 100, 200, and 400 kHz frequencies, respectively (Figure 6a). For context, the average range of backscatter values in a local 3 × 3 cell neighborhood (i.e., the "speckle") for the 2016 target datasets in the test area were 2.6, 2.6, and 3.5 dB, for 100, 200, and 400 kHz layers. The most complex method (the BRT(back*bath) model with interaction) produced the lowest fitted MAE of all methods for all frequencies, but never achieved the lowest test MAE. It consistently overfit the data (see the difference between fitted and test MAE, $\theta$; Figure 6a), producing over-optimistic MAE estimates. SLR(back) and BRT(back) produced slightly over-optimistic fitted MAEs, as did BRT(back+bath). The mean, SLR(bath), and MLR methods each produced fitted MAE values that were conservative when compared to the test values.

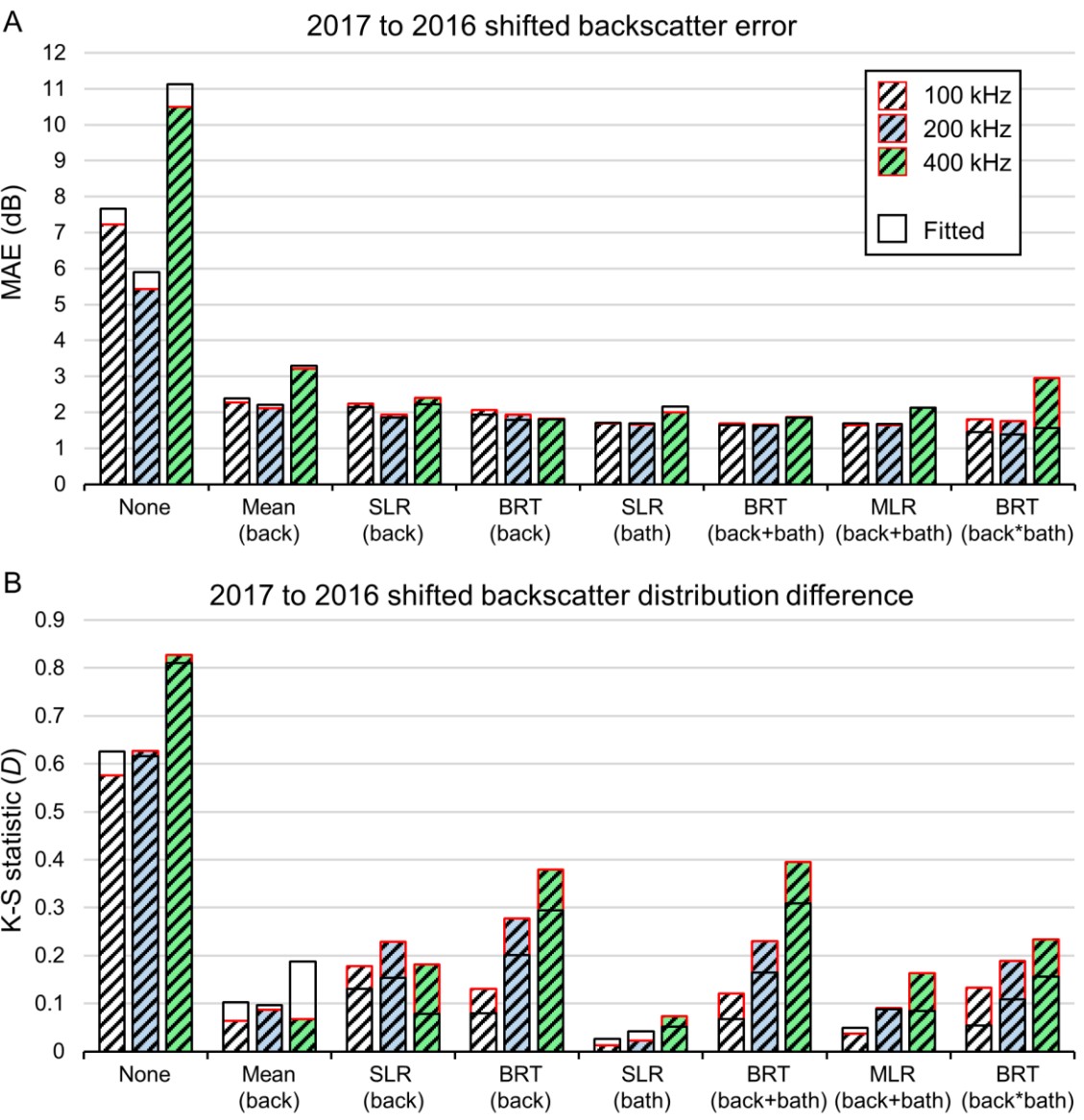

**Figure 6.** (**a**) Fitted and test mean absolute errors (MAEs) of 2017 Bedford Basin data corrected to 2016 for 100, 200, and 400 kHz frequencies with seven bulk shift methods, and (**b**) Kolmogorov–Smirnov (K-S) statistics for distributions of 2016 Bedford Basin data compared with shifted 2017 frequencies.

**Table 3.** Performance comparison of bulk shift methods at harmonizing Bedford Basin datasets from different years. Scores are average ranks relative to the other methods for each frequency. Ranks are colour-coded from blue to orange for best to worst.

| Method | MAE Rank | $D$ Rank | $\theta_{MAE}$ Rank | $\theta_D$ Rank |
|---|---|---|---|---|
| Mean | 7.00 | 2.00 | 1.33 | 1.33 |
| SLR(back) | 5.67 | 5.33 | 5.33 | 5.33 |
| SLR(bath) | 3.00 | 1.33 | 2.33 | 2.00 |
| MLR | 2.00 | 2.67 | 2.33 | 2.67 |
| BRT(back) | 3.67 | 6.00 | 5.67 | 5.33 |
| BRT(back+bath) | 2.00 | 5.67 | 4.00 | 5.33 |
| BRT(back*bath) | 4.67 | 5.00 | 7.00 | 6.00 |

K-S statistics ($D$) suggested that SLR(bath) produced data distributions that best matched the target backscatter dataset, followed by the mean backscatter shift and MLR (Table 3). These three methods also produced conservative $D$ values (i.e., low values of $\theta_D$; Figure 6b; Table 3). The BRT methods generally produced higher values of $D$, indicating shifted data distributions that are dissimilar to those of the target datasets.

The simulated mosaic and error maps show that the mean backscatter-shifted corrections performed well at the mid-dB range of the data, but poorly at the extremes for 100, 200, and 400 kHz 2017 data corrected to 2016 (e.g., Figure 7, Figure 8). SLR(back) performed better at high and low backscatter values, but the mosaics suggest much of the mid-range was poorly matched with extensive non-uniform error. BRT(back) produced a "washed out" appearance, where the dynamic range of the 2017 data was compressed and fine spectral details lost, producing patchy error (see Appendix B for an explanation of dynamic range compression). SLR(bath) produced mosaics that were hardly distinguishable from the original 2016 data, with error uniformly distributed throughout the test area. BRT(back+bath) produced similar results to BRT(back), with patchy error and a loss of spectral detail. MLR produced mosaics that looked very close to the target 2016 data with mostly uniform error, similar to SLR(bath) (e.g., Figure 7, Figure 8). BRT(back*bath) consistently resulted in a loss of spectral detail and a mosaic markedly different from the 2016 reference, with generally non-uniform error. Appendix C presents the highest quality mosaics for each simulation.

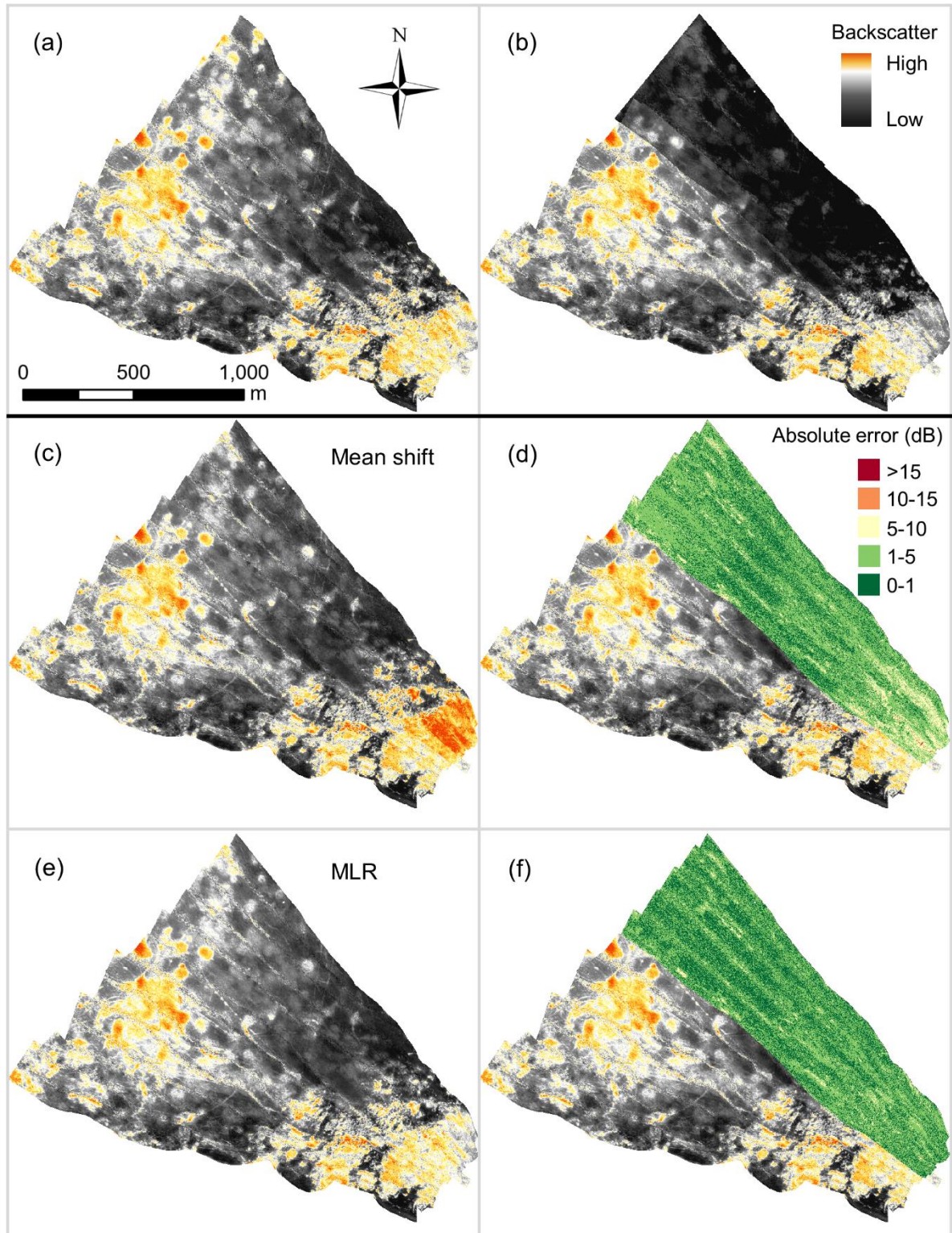

**Figure 7.** (**a**) Target 2016 100 kHz Bedford Basin backscatter dataset, (**b**) uncorrected mosaic with 2017 100 kHz data, (**c**) mean-shifted backscatter mosaic, (**d**) spatial distribution of mean-shifted backscatter error, (**e**) MLR-shifted mosaic, and (**f**) spatial distribution of MLR-shifted backscatter error.

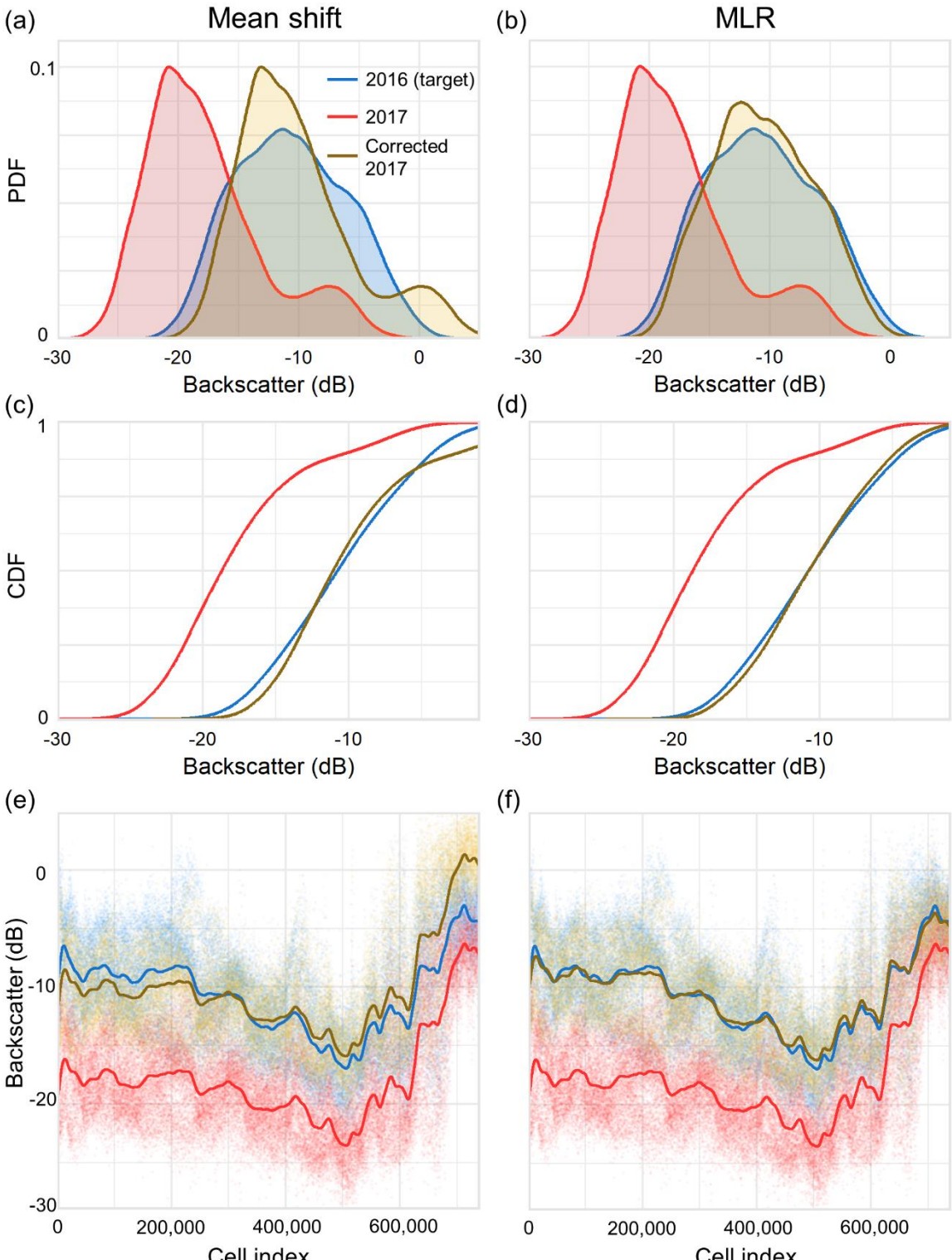

**Figure 8.** Bedford Basin inter-year 100 kHz test data comparison. Probability density functions (PDFs) of (**a**) mean backscatter, and (**b**) MLR shifts; cumulative distribution functions (CDFs) of (**c**) mean backscatter, and (**d**) MLR shifts; distribution of backscatter values with first order LOESS curves for (**e**) mean backscatter, and (**f**) MLR shifts, plotted by cell index on the *x*-axis, in which the rasters are read (north to south).

### 3.2. Harmonizing Different Frequencies

The error between target and corrected shift datasets that comprise multi-frequency mosaics was generally related to the similarity between the frequencies being combined, whereas similarities between the data distributions were not always clearly linked to frequency. The MAE of mosaics comprising 100 kHz layers shifted to 200 kHz was nearly always lower than when shifted to 400 kHz (Figure 9). Similarly, mosaics of 400 kHz layers shifted to 200 kHz were nearly always more accurate than when shifted to 100 kHz. Neither 200 kHz shifted to 100 or to 400 kHz had consistently lower errors. Data distributions were consistently more similar between the corrected 100 kHz and 200 kHz datasets than 100 and 400 kHz (Figure 10), yet the distributions of corrected 400 kHz data were not always closer to 200 kHz when compared to 100 kHz in Patricia Bay.

The lowest MAE values that were achieved for multi-frequency shifts between 100 and 200, or 200 and 400 kHz, were between 1.4–1.7 dB, and were between 1.5–2.3 dB for 100 and 400 kHz. For context, the average range of backscatter values in a local $3 \times 3$ cell neighborhood for 100, 200, and 400 kHz target layers in the test area were 2.6, 2.6, and 3.5 dB for the Bedford Basin, and 2.5, 2.6, and 2.9 dB for Patricia Bay, respectively. BRT(back+bath) generally produced mosaics with the lowest MAE for Bedford Basin and Patricia Bay datasets, followed by MLR, and SLR(bath) (Table 4). Despite their accuracy, the MAE of the fitted BRT models was often higher than the test values, which suggested that these methods might be prone to overfitting. The fitted MAE values were closer to the test for MLR and SLR(bath). BRT(back*bath) always had the lowest fitted MAE, yet this was never corroborated by the test values, which were always higher (Figure 9). This indicated that the model fit statistics were not a reliable indicator of the quality of BRT(back*bath) corrections. The MAE values of the mean backscatter error shifts were closest to the test values, yet these were higher than the other methods. In general, $\theta_{\mathrm{MAE}}$ increased with the complexity of the model (Table 4).

The K-S tests suggested that SLR(bath) produced mosaics with data distributions that were the closest to the target dataset, and that these were generally well reflected by the fitted model results (Figure 10). The mean backscatter error shifts and MLR produced data distributions that were similarly close to those of the target data, yet estimates of $D$ from the MLR fitted models were often over-optimistic, while the mean shifts were closer to the actual test values (Table 4). Of the non-parametric methods, BRT(back+bath) generally produced distributions with the lowest value of $D$, but high values of $\theta_D$, suggesting overfitting.

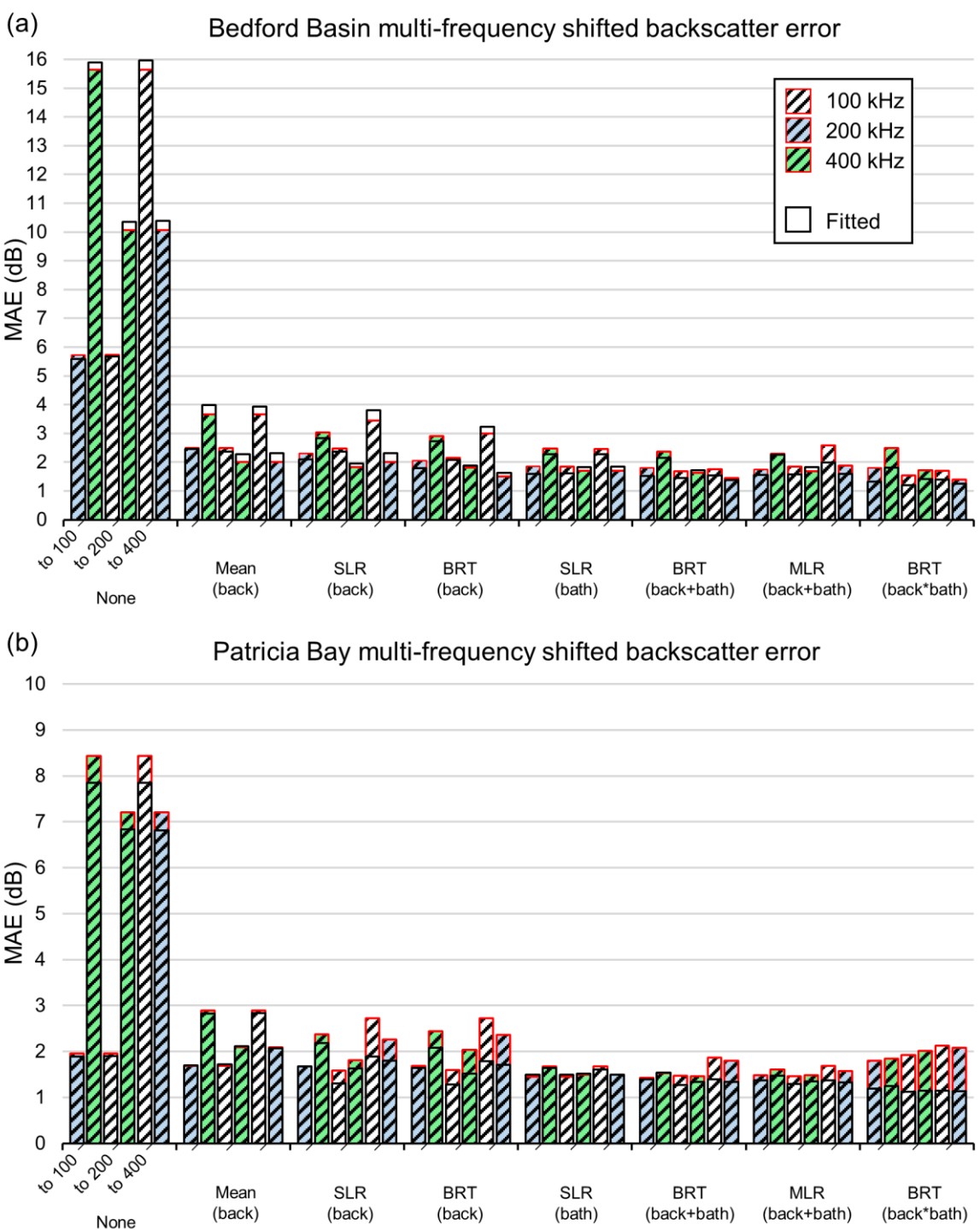

**Figure 9.** Fitted and test MAE of 100, 200, and 400 kHz backscatter corrected to other frequencies using seven bulk shift methods for (**a**) Bedford Basin and (**b**) Patricia Bay datasets.

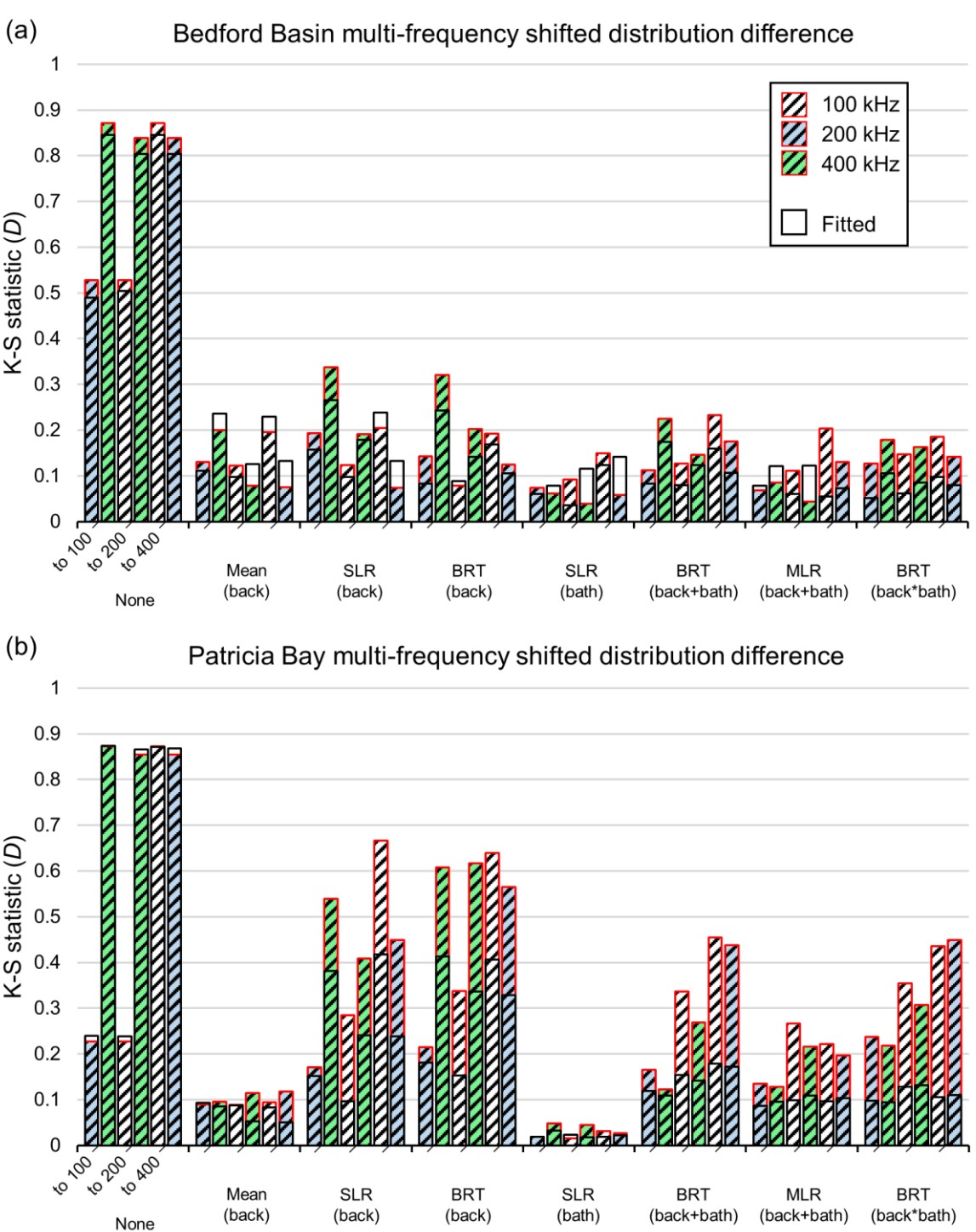

**Figure 10.** K-S statistic for shifted distributions of 100, 200, and 400 kHz backscatter corrected to other frequencies using seven bulk shift methods for (**a**) Bedford Basin and (**b**) Patricia Bay datasets.

**Table 4.** Performance comparison of bulk shift methods at harmonizing datasets of different frequency for Bedford Basin and Patricia Bay datasets. Scores are average ranks for all frequencies and both datasets, relative to the other methods. Ranks are colour-coded from blue to orange for best to worst.

| Method | MAE Rank | D Rank | $\theta_{MAE}$ Rank | $\theta_D$ Rank |
|---|---|---|---|---|
| Mean | 6.58 | 2.92 | 2.04 | 2.00 |
| SLR(back) | 5.42 | 5.54 | 3.38 | 3.92 |
| SLR(bath) | 2.58 | 1.17 | 3.29 | 2.54 |
| MLR | 2.50 | 3.08 | 3.83 | 3.92 |
| BRT(back) | 5.25 | 5.50 | 4.29 | 4.71 |
| BRT(back+bath) | 1.92 | 4.75 | 4.33 | 4.42 |
| BRT(back*bath) | 3.75 | 5.04 | 6.83 | 6.50 |

Bulk shift and error distribution maps suggested that the K-S statistic ($D$) was generally indicative of mosaic quality. SLR(bath)—the method with the lowest value of $D$ on average—produced mosaics that were comparable to the target (e.g., Figure 11), with highly similar data distributions (e.g., Figure 12). Although BRT(back+bath) produced corrections with the lowest MAE values on average, the mosaics were not necessarily of the highest quality (e.g., Figure 11, Figure 12), demonstrating the "washed out" effect noted earlier (and discussed in Appendix B). This was reflected by the values of $D$, which were generally higher than the simpler methods (Table 4), indicating that data distributions were dissimilar to those of the target dataset (e.g., Figure 12). This is well-illustrated by the PDF, which shows that the distribution of 200 kHz backscatter values has been truncated at the extremes and replaced with an over-representation of moderate values between −16 and −20 dB (Figure 12a). Likewise, SLR(back) and BRT(back) had high values of $D$ and they generally produced poor mosaics. The mean backscatter error shifts generally produced low values of $D$, which were similar to the fitted values, but the highest MAE values. Correspondingly, though the dynamic ranges of values in mean-shifted mosaics often closely matched those of the target dataset, they contained areas with large amounts of error, producing high MAE values. In other words, the corrected portions of mean-shifted mosaics had a dynamic range that was similar to the target dataset, but the spatial distribution of those values was poorly matched, as evidenced by non-uniform distributions of error and low MAE statistics relative to the other methods (Table 4). Appendix C presents the highest quality mosaics for each simulation.

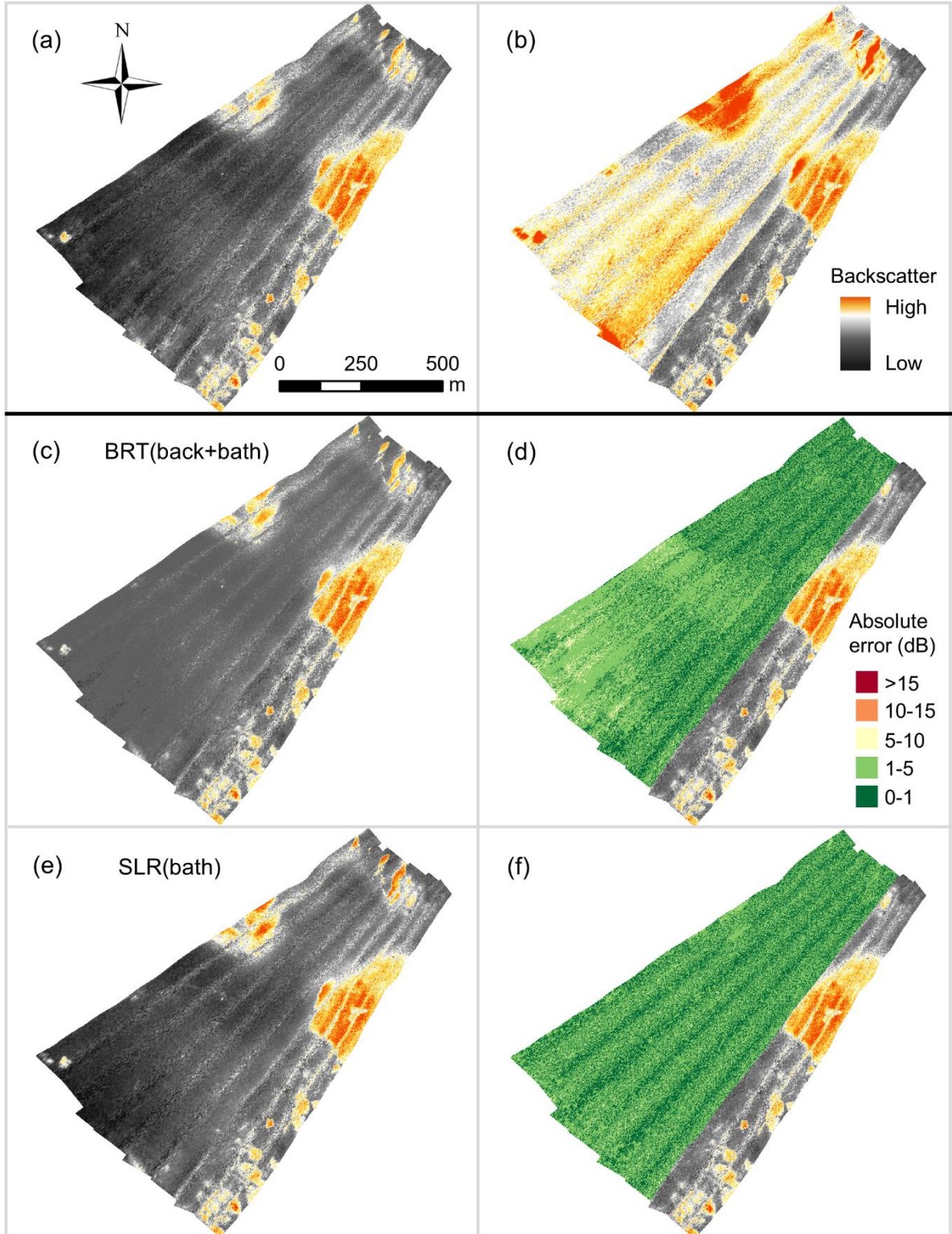

**Figure 11.** (**a**) Target 400 kHz Patricia Bay backscatter dataset, (**b**) uncorrected mosaic with 200 kHz data, (**c**) BRT(back+bath) mosaic, (**d**) spatial distribution of BRT(back+bath)-shifted backscatter error, (**e**) SLR(bath) mosaic, and (**f**) spatial distribution of SLR(bath)-shifted backscatter error.

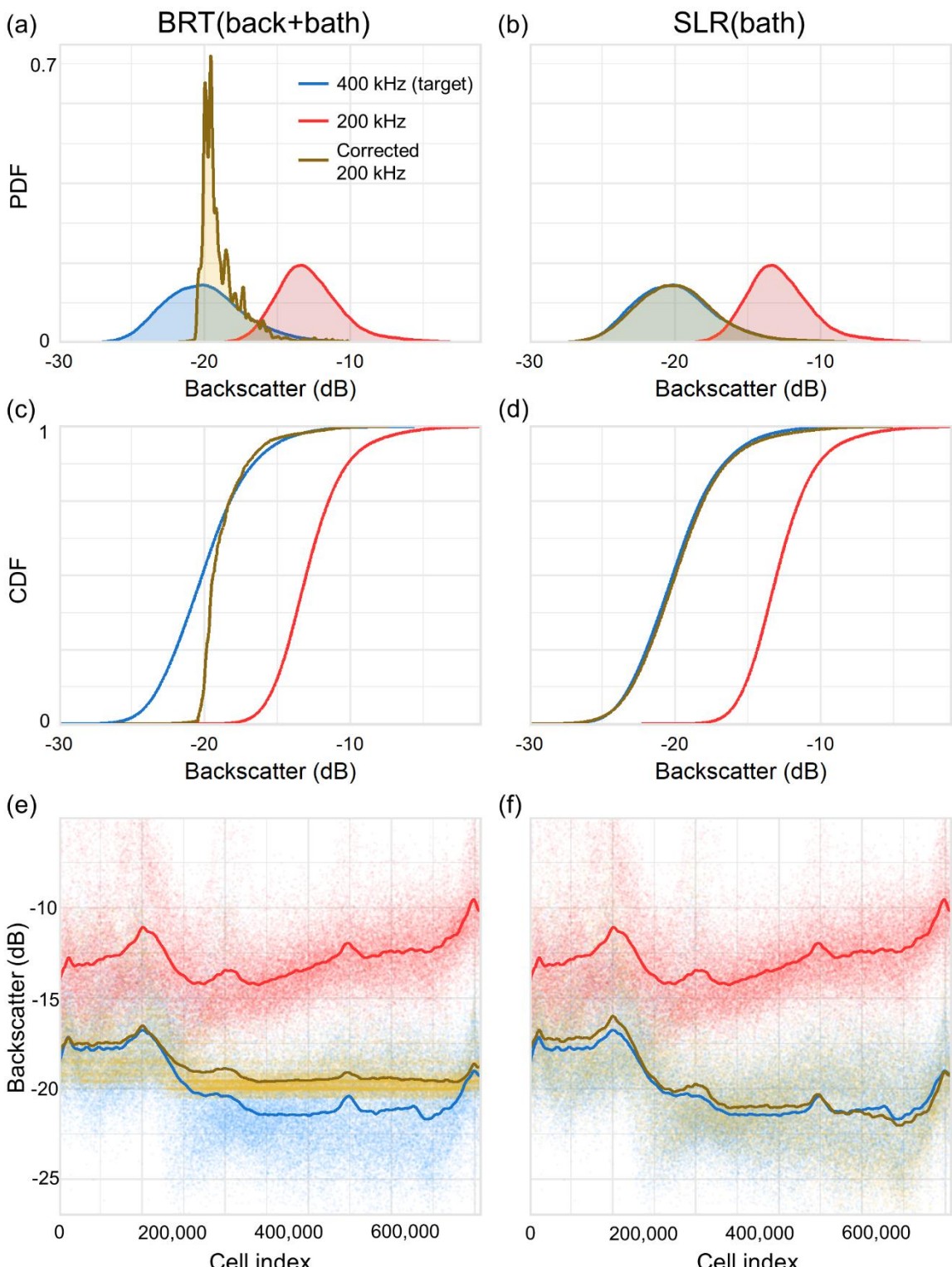

**Figure 12.** Patricia Bay 200 and 400 kHz test data comparison. PDFs of (**a**) BRT(back+bath), and (**b**) SLR(bath) shifts; CDFs of (**c**) BRT(back+bath), and (**d**) SLR(bath) shifts; distribution of backscatter values with first order LOESS curves for (**e**) BRT(back+bath), and (**f**) SLR(bath) shifts, plotted by cell index on the *x*-axis, in which the rasters are read (north to south).

## 4. Discussion

It is common for seabed mappers to inherit uncalibrated MBES data from several surveys, often in a processed form and without raw data. Ideally, backscatter acquisition should be calibrated to facilitate downstream comparison (e.g., using calibration targets or natural reference areas; [20,22]), yet it is impossible to ensure that these practices are universally adopted. The need to establish methods for efficiently handling uncalibrated multi-source backscatter datasets will persist even if calibration is eventually widely adopted, given the large amount of pre-existing MBES data. In a geological or biological mapping context, independent analysis of the datasets and *post hoc* combination of results is an effective solution when the number of backscatter datasets is relatively low, and each has been adequately ground-truthed [26], but this becomes problematic when ground-truth locations are unevenly or sparsely distributed over the backscatter layers. Multi-source backscatter harmonization can address some of these issues, but it has previously been performed ad hoc (e.g., [9,25,28]). Here, we present the first investigation, to our knowledge, into standardized and repeatable methods for multi-source backscatter harmonization for seabed mapping. Although we focus on application to MBES data here, these methods are potentially applicable to backscatter data of other sources, such as side-scan sonar, including interferometric and synthetic aperture systems.

The results suggest that the harmonization of backscatter data acquired by a single MBES system during separate surveys is generally feasible using conservative bulk shift approaches. Here, the accuracies of bulk shifts using multiple linear regression were at least comparable, and often superior, to the most complex methods tested (BRTs). Furthermore, multiple linear regression largely avoided issues associated with model overfitting; the fitted model statistics were generally representative of the quality of the bulk shift in the test area. More flexible and complex models sometimes performed well, but nearly always overfit the data, producing fitted evaluation statistics that were over-optimistic regarding the error and distributions of the shifted data. The backscatter datasets used for these simulations were collected one year apart; greater temporal variation might be expected between legacy datasets that comprise longer time periods as a function of changes to the surficial geology of the seafloor.

The results suggest that harmonizing backscatter datasets of multiple frequencies is also feasible, with the caveat that success will partly depend on the extent to which the frequencies differ. In our results, harmonized mosaics were generally of higher quality when frequencies were more similar. Mosaics comprising datasets of different frequency produced using multiple regression were comparable, and sometimes superior, to those produced using the same frequency but from separate surveys (e.g., 200 to 400 kHz in Figure 9 vs. 400 kHz in 2017 to 2016 in Figure 6). Conversely, harmonized mosaics from 100 and 400 kHz frequencies were never as good as those produced using the same frequency from different surveys.

The frequency-dependent response of the seafloor is potentially the greatest challenge when attempting to harmonize backscatter datasets that were acquired using multiple MBES operating frequencies. It is well-accepted that the acoustic response of the seafloor is a function of the operating frequency of the MBES system [33–37]. Frequency-dependent interactions between an acoustic signal and the seafloor (e.g., penetration depth) produce specific backscatter responses, which may be partially or fully lost at different frequencies when applying harmonization approaches [25]. We may never expect to fully match the spectral detail between datasets of disparate operating frequency at a high resolution; these simulations suggest that the feasibility of harmonizing datasets of different frequency depends on the extent to which the frequencies differ, and the site-specific characteristics of the surficial substrata (e.g., stratification within the acoustic penetration depth, as observed at the Bedford Basin; see [15]).

Relatedly, we expect data resolution to be an important factor in determining the feasibility of harmonizing multi-source backscatter datasets. MBES bathymetric and topographic error is partially dependent on spatial scale [38], and we expect the same to be true for backscatter. The expected result of coarsening backscatter data resolution is a reduction in both natural variability (i.e., "noise") and

the ability to resolve fine scale seabed heterogeneity—both of which may facilitate harmonization. Different operating frequencies may detect unique fine-scale features at a high resolution, but they may describe broader, more general substrate trends at coarser resolutions. This is a form of scale dependence that might be exploited to render multi-frequency datasets comparable, depending on the intended use of the backscatter mosaic.

The qualities of the bulk shift methods examined in this study were partly evaluated on the representativeness of the fitted evaluation statistics compared to actual performance (i.e., "overfitting"). Normally, the fitted statistics from where backscatter datasets overlap would be the only information available to the operator. Therefore, it is important to determine how well these represent the actual quality of the correction, and if a given method is prone to overfitting. In some instances, it might be possible to design an independent assessment, but it would need to be spatially explicit (e.g., spatial blocking; [39]), since the large amount of spatial autocorrelation within a continuously-sampled dataset hinders other common assessment techniques, such as cross-validation. An adequate blocking design might be difficult to achieve when the overlap between datasets is minimal, or where large spatial blocks would reduce the range of sampled values. The latter could result in sub-optimal bulk shift corrections where there are non-uniform backscatter differences in environmental, and by association, geographic space.

Visual assessment remains an important component to determining the quality of multi-source backscatter mosaics, and the operator should be attentive to several indicators of poor harmonization. Visual artefacts in the final mosaic along the boundaries between datasets are the most obvious sign that they have been poorly integrated. These may occur at only certain areas of the boundary, which likely means that the difference between backscatter datasets is non-uniform across either the dynamic range of the datasets or other environmental variables. Potential solutions are to explore the integration of auxiliary variables, such as bathymetry, or to adopt a non-linear modelling method—although the results here suggest that the latter should be approached cautiously. Apparent differences in spectral qualities, such as the dynamic range between sections of the mosaic, should be scrutinized. A "washed out" effect in the corrected portion of the mosaic (e.g., Figure 11c) might suggest that the shifted dataset had a greater dynamic range than the target in the area where they overlapped. Our results suggest that this effect is more likely to occur with flexible modelling methods, which should be avoided in such cases.

Additional covariates other than the backscatter values of the dataset being corrected may be used to improve the quality of bulk shifts in some cases. The degree to which bathymetry explained the error between backscatter datasets for both sites in this study was surprising. In all cases, there was a linear increase in absolute backscatter error with depth for datasets of different operating frequency (Figure 13). At both study sites, the slope of the linear relationship corresponded to the magnitude of difference between operating frequencies (i.e., steeper slope for 100 to 400 than 200 to 400 kHz shifts). It was particularly interesting that the error between 100 and 200 kHz datasets was near zero at <20 m for both sites, but increased substantially with depth. Initially, the possibility that this was caused by a correlation between depth and substrate type (and frequency-dependent response) seemed likely, but the gradation and linearity of the relationship motivated an alternative explanation. The linearity was highly similar for each combination of frequencies at both sites and persisted even when a consistent relationship between backscatter and depth was lacking, suggesting that it was not caused solely by a correlation of substrate with depth. The sign of the slope for this relationship corresponded to the order in which frequencies were harmonized at both sites. Where a lower frequency was corrected to a higher one, greater positive error with depth was observed; where a higher frequency was corrected to a lower, greater negative error with depth was observed. This suggests that the corrected relative intensity of higher frequencies decreased at a greater rate than the lower frequencies with increasing depth.

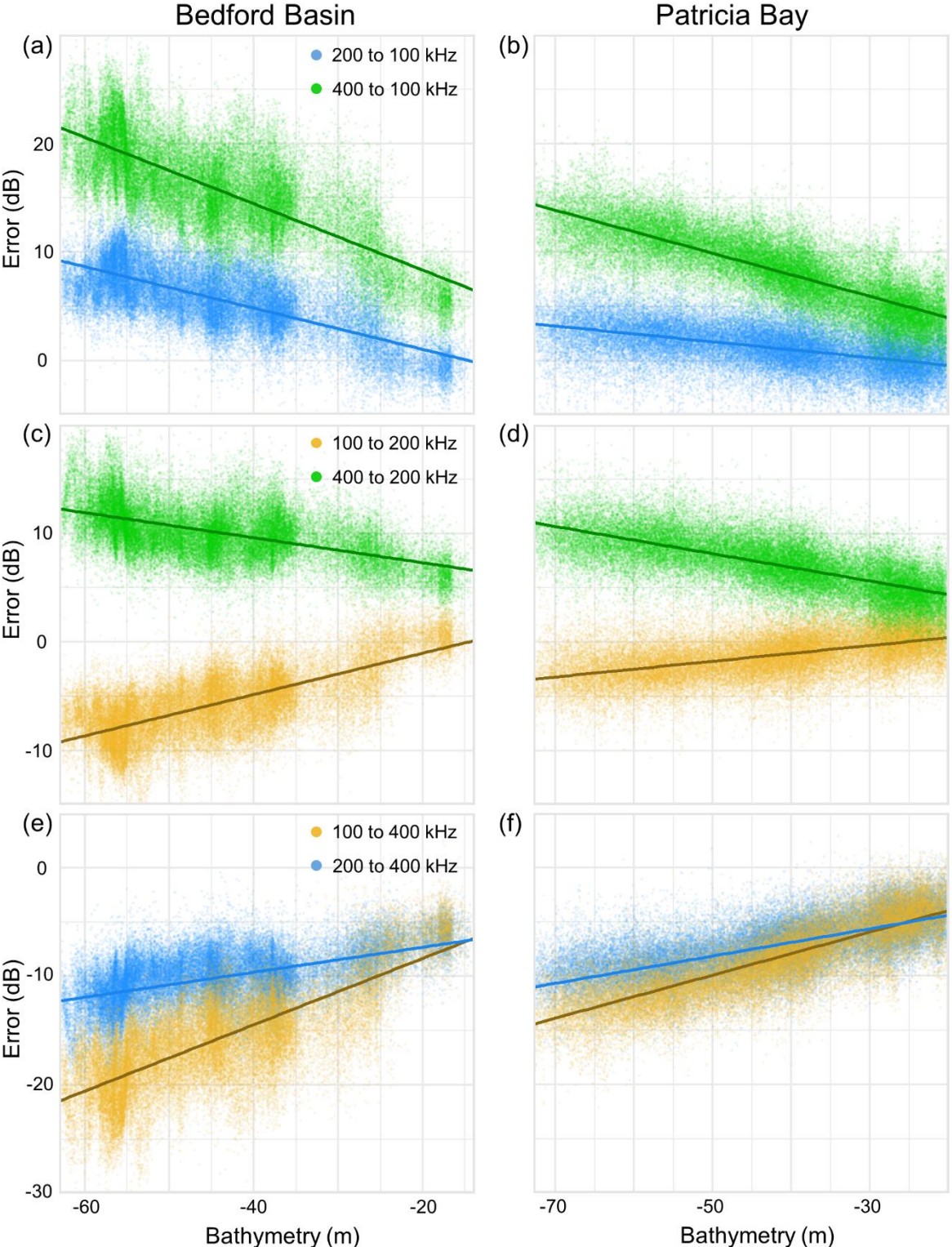

**Figure 13.** Linear relationships between bathymetry and backscatter error for multiple frequencies at (**a**), (**c**), (**e**) Bedford Basin, and (**b**), (**d**), and (**f**) Patricia Bay.

We suggest two potential explanations for the linear change of backscatter error (i.e., difference) with depth at these sites. It would be most simple to conclude that the calculation of transmission loss (TL) was systematically incorrect because of errors in the absorption coefficient calculation—possible, for example, if the frequency-dependent decrease in attenuation with depth was modelled incorrectly.

There is some evidence to support this theory, as there were also consistent linear increases in positive error with depth between datasets of the same frequency collected over multiple years at the Bedford Basin. In other words, the intensity of the 2017 data became relatively weaker than the 2016 with increasing depth, for all frequencies. The magnitude of increase in error with depth between frequencies would be surprising, though, at a depth range of only ~50 m if it was due to a depth-attenuation calculation error (see [1]). Alternatively, the increase of error with depth might be at least partially characteristic of how the signal from the R2Sonic 2026 interfaces with the substrate, given the depth, for each frequency. For example, the increase in error with depth might be a function of frequency-dependent substrate penetration and bottom loss (i.e., sediment attenuation [33]). As depth increases, the intensity of an acoustic signal attenuates due to absorption throughout the water column, but at a greater rate for higher frequencies [1]. This is accounted for in post-processing to produce backscatter values that are depth independent. However, a factor that cannot be easily accounted for is how bottom loss (L; [36]) might co-depend on frequency and depth, given decreased energy at the seafloor due to water column attenuation. This problem becomes increasingly complex while considering the acoustic properties of different substrate types, and the potential for differences in acquisition parameters. Regardless, it seems that the linear relationship between depth and error describes a consistent difference in how each frequency represents the substrate, given water depth.

From a survey at the International Marine Geological and Biological Habitat Mapping conference (GeoHab) in 2014, Lucieer et al. [12] identified "lack of calibration required for optimizing backscatter data", "the lack of standardized methods available for referencing", and "the ongoing struggles with large data volumes" as the top concerns among backscatter end-users. Therefore, the work presented here is part of a larger effort to develop repeatable and standardized approaches for the acquisition and use of backscatter data for seabed biological and geological mapping (e.g., [13,40]. This effort has become increasingly urgent given the proliferation of MBES as a seabed mapping tool and the importance of backscatter as a proxy for substrate properties. Though the principles of underwater acoustics and sonar for seafloor mapping are well established [1,13,36], the rapid development of MBES technology and its widespread uptake by a diversity of disciplines has created a knowledge gap regarding how to best implement acoustic data for biological and geological applications [12]. Exciting new developments such as multispectral MBES continue to advance the power of acoustics for these applications, yet the advance of knowledge regarding how to utilize these technologies appropriately needs to keep pace. This is a great challenge, but its importance cannot be overstated given the urgent need for spatial information on marine ecosystems.

## 5. Conclusions

Our findings suggest that relatively straightforward statistical methods were effective at harmonizing backscatter datasets of the same frequency from separate surveys. We suggest that simple parametric approaches may often be preferable to more flexible ones based on the error and distributional statistics presented here, and the quality of the simulated mosaics. Multiple linear regression facilitates interpretation of multiple predictors, while retaining the distribution of the data–largely avoiding issues such as dynamic range compression and overfitting. More flexible approaches in this study produced unnecessarily complex corrections that altered the distribution of backscatter values in the dataset being shifted, to the detriment of mosaic quality. It is possible that flexible machine learning approaches, such as BRTs, may be better suited to instances of extensive overlap between datasets, where the entire dynamic range of the backscatter dataset undergoing correction has also been covered by the target layer. This has not been tested in detail, and requires further investigation.

Harmonizing backscatter datasets of different frequency is feasible, but whether it is appropriate depends on the magnitude of difference between frequencies. Therefore, critical to the harmonization procedure is the operator's ability to assess the mosaic quality, and the tools available for this task are the fitted model statistics and the mosaic map. Although not explored in detail here, we also expect

the data resolution to be an important factor in determining whether multi-frequency harmonization is feasible. Here, flexible modelling methods sometimes produced accurate mosaics using datasets of different frequency, but the fitted statistics were generally over-optimistic. Based on these findings, we advise caution in applying highly flexible non-parametric models for backscatter harmonization, especially when the overlap between datasets is limited. Again, simpler regression methods can largely avoid issues that are associated with overfitting. Visual analysis of the harmonized mosaic and the data distributions can serve as indicators of the quality of the harmonized product, alongside analysis of the fitted statistics.

It would be highly beneficial to the harmonization procedure if the acquisition of new backscatter data within an area of previous surveys overlapped a representative proportion of benthic conditions that were previously mapped. This could be efficiently accomplished by ensuring that the new survey covers at least the entire backscatter dynamic range of the previous survey, and ideally also the bathymetric range. We suggest that this is more efficient than maximizing the size of the overlapping area for harmonization. Although both are desirable, the former should be prioritized if necessary.

**Supplementary Materials:** The following are available online at http://www.mdpi.com/2072-4292/12/4/601/s1, Code S1: bulkshift.R, Document S1: bulkshift.R Function Tutorial. The multispectral MBES datasets are available upon request from the authors or R2Sonic.

**Author Contributions:** Author contribution to this paper is as follows. Conceptualization, B.M., C.J.B., M.L.; methodology, B.M.; software, B.M.; validation, B.M., K.R.; formal analysis, B.M.; investigation, B.M.; resources, C.J.B., K.R.; data curation, C.J.B., B.M.; writing—original draft preparation, B.M., C.J.B.; writing—review and editing, B.M., C.J.B., K.R., M.L.; visualization, B.M.; supervision, C.J.B., K.R.; project administration, B.M.; funding acquisition, K.R., C.J.B. All authors have read and agreed to the published version of the manuscript.

**Funding:** B.M. was supported by post-doctoral funding through a Canada Research Chair in Ocean Mapping and the Department of Fisheries and Oceans Freshwater Science Contribution Program. Data acquisition was supported by NSERC grant CIRC472115-14.

**Acknowledgments:** Thank you to R2Sonic for providing the multispectral data used in this study and Quality Positioning Services (QPS) for software support.

**Conflicts of Interest:** The authors declare no conflict of interest.

## Appendix A

*Selected Modelling Methods*

A limited selection of modelling methods were tested in this study to broadly span the rigid and parametric, up to flexible and non-parametric. There are many other potential methods, including a diverse selection of flexible models. It is worth briefly noting the appropriateness of some characteristics that make a given modelling method potentially suitable for bulk shift applications.

Linear regression performed well in these simulations, and a natural extension would be to test more flexible regression approaches. How these may translate to corrections outside the range of "sampled" (i.e., overlapping) backscatter intensities, though, has important consequences for the harmonized mosaic. When the backscatter layer undergoing correction contains values that are far outside of those being modelled (from the area of overlap), the extrapolated model values may become extreme, for example, when the model contains polynomial curves. These can produce corrections that are far beyond the original dynamic range of the dataset.

Among flexible non-parametric models, BRTs behave differently than many other approaches at predicting beyond the range of sampled values. BRTs, like random forest, are based on classification and regression trees (CART), which use recursive data partitioning to model the response as a function of one or more predictors. These tree-based methods are not, therefore, truly continuous; their predictions comprise many small, discrete segments corresponding to the response variable partitions. The effect of such approaches in the context of bulk shift correction is that the extremes of the fitted (sampled) backscatter intensities, from the area of survey overlap, are "clamped", wherein all predictions outside of the range of fitted values are assigned the same prediction as the most extreme fitted values.

Conversely, methods such as artificial neural networks and generalized additive models are truly continuous (and smooth), and the predicted trends at the ends of the fitted data will be extrapolated to new data. This is potentially desirable since we may expect new data to follow previously established trends, but it also makes extrapolation sensitive to the extremes of the fitted data, which could result in erroneous values in the harmonized mosaic.

Like most modelling applications it is ideal to strike a balance between flexibility and overfitting, but in this context it is especially important to keep in mind the effects of extrapolation. While this balance may be possible to achieve by tuning hyperparameters of flexible machine learning models, the success in doing so will depend on the application, and it is difficult to recommend such an approach generally. Intermediate alternatives might be appropriate though. For example, multivariate adaptive regression splines (MARS) using a low number of hinge functions (e.g., 1 or 2; see Supplementary Document S1) would be very similar to linear regression, but could better fit datasets where the slope of the regression varies according to the dB level, and/or another covariate (e.g., bathymetry). Alternatively, it may be the case that flexible modelling approaches are more preferable when there are large amounts of overlap between the backscatter datasets being harmonized, in which the entire dynamic range of the dataset undergoing correction has also been covered by the target dataset. Such cases may avoid the potential pitfalls associated with extrapolating corrections beyond the range of the fitted data.

## Appendix B

*Dynamic Range Compression*

When a range of backscatter values in the layer undergoing a bulk shift correction (the "shift" layer) correspond to a narrower range of values in the target layer, there is a risk of losing spectral detail in the harmonized mosaic. This is demonstrated in Figure A1 using two datasets from this study.

The compression of dynamic range is most likely to occur using non-parametric flexible modelling methods, such as boosted regression trees, random forest, artificial neural networks, and generalized additive models, which all have the ability to assign multiple values of the response variable (the shift layer here) to a single value of a predictor variable (the target layer). While this may be desirable in many modelling applications, it can be precarious in a bulk shift context. Figure A1 demonstrates how overfitting a certain portion of the shift dataset using BRTs can cause a range of intensity values to be assigned a much narrow range in the corrected layer. The linearity of the shift layer has also been lost in the BRT corrected layer, which manifests in the backscatter mosaic as a loss of spectral detail.

Additionally, if the intensity values of two backscatter datasets differ across another predictor variable that has not been included in the model, such as bathymetry, multiple intensity values of the shift layer may be observed at a single value of the target layer, which are dependent on the depth that is unaccounted for. This may cause a range of intensity values of the shift layer to be corrected to the same value, regardless of depth. Dynamic range compression could also be caused by temporal variation between datasets. If changes to the seabed substrate have occurred over time, new intensity values at certain locations will correspond to estimates of error at those locations, rather than error that is systematic throughout the dataset. If the statistical relationship between the shift and target datasets is conflated with the temporal error at certain locations, flexible models may incorrectly apply them to the remainder of the shift dataset. This could cause the correction of multiple values in the shift layer to fewer in the target layer, compressing the range of intensity.

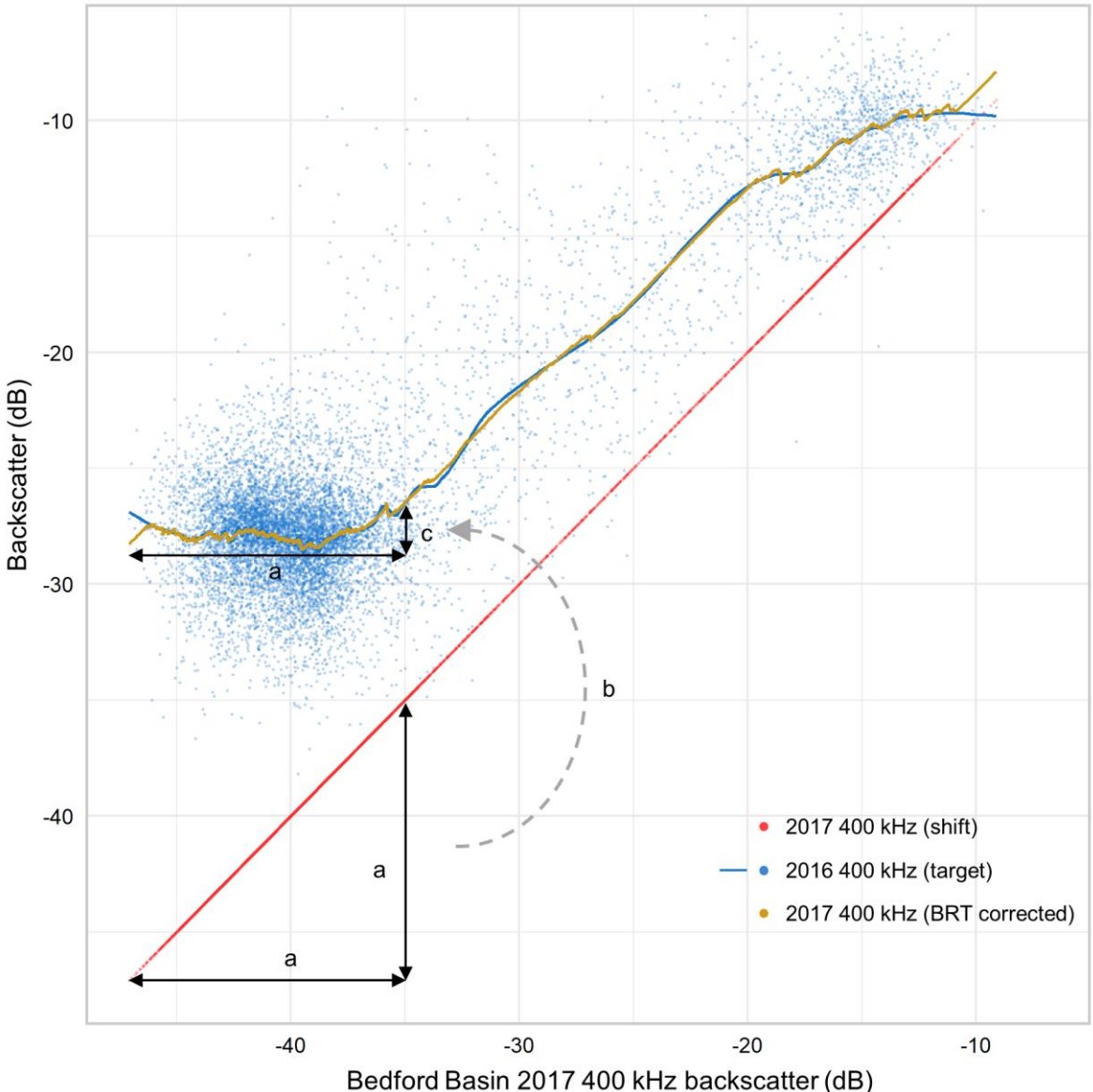

**Figure A1.** 2017 400 kHz Bedford Basin backscatter data on the *x*-axis, against 2017 (1:1 plot), 2016 (with LOESS curve), and BRT-corrected 2017 on the *y*-axis. Arrows delimit (**a**) the original ~12.5 dB range of backscatter values observed in the 2017 dataset, that were (**b**) compressed by the BRT bulk shift, to (**c**) a ~2 dB range based on values observed in the 2016 dataset.

More rigid or parametric methods are less likely to suffer the effects of dynamic range compression. Specifically, methods such as linear models are more robust to localized or inconsistent relationships between backscatter datasets, and generally avoid correcting multiple shift layer intensity values to a single target value. Any method that derives systematic statistical relationships over the entire dynamic range of the shift dataset should, therefore, be fairly robust.

**Appendix C**

*Bulk Shift Simulation Mosaics*

Presented here are the highest quality mosaics for each simulation in this study. The quality was determined by the ranks of MAE and *D* evaluation statistics. Each figure corresponds to bulk shift simulations for a given frequency and study site. The top left pane of each is the target layer. The top

right panes for the Bedford Basin datasets are the 2017 data of the same frequency, corrected to the 2016 data. Bottom panes for all figures are multi-frequency bulk shift mosaics.

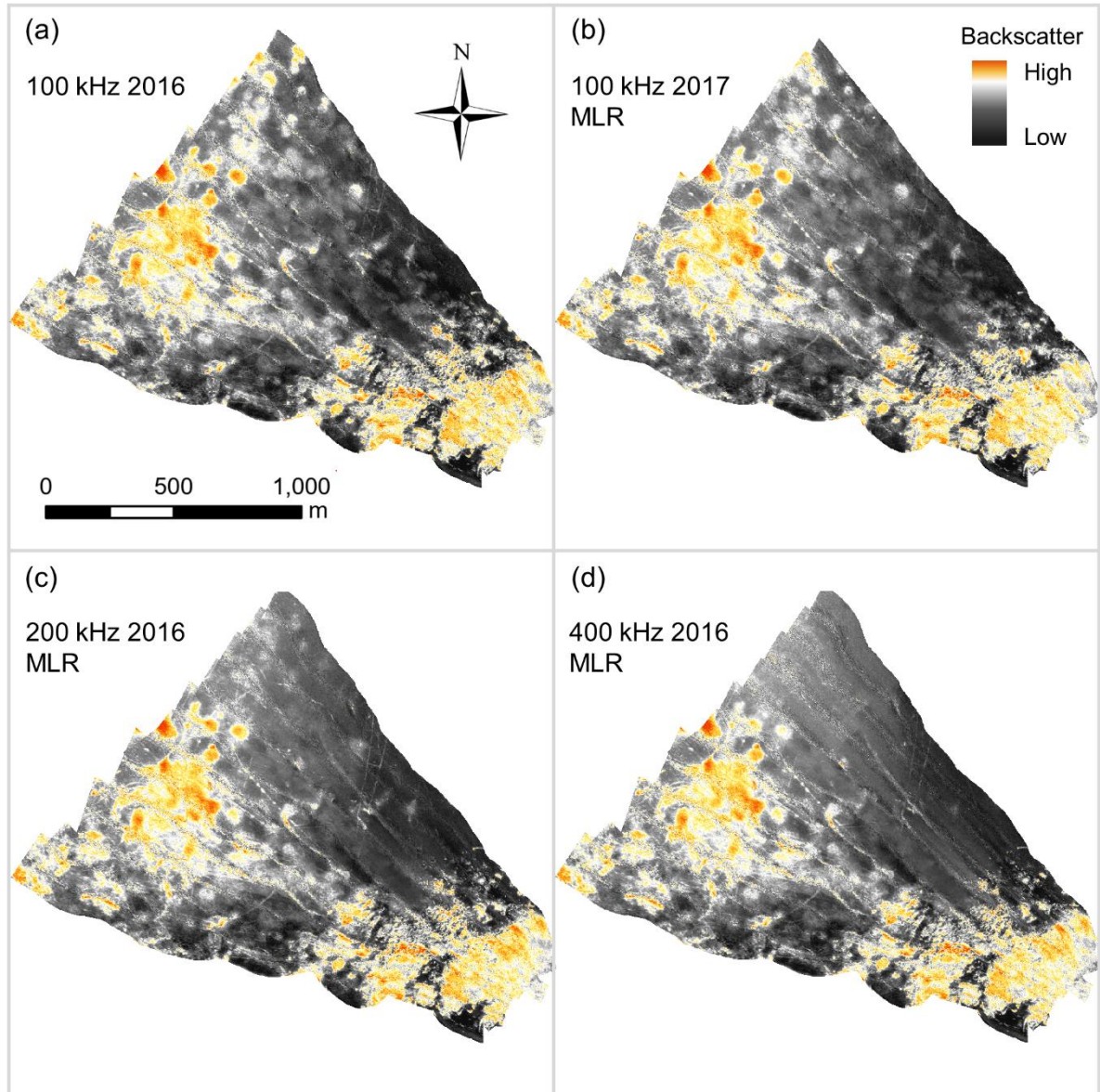

**Figure A2.** Highest quality bulk shift mosaics of (**a**) the 2016 Bedford Basin 100 kHz dataset, with (**b**) 2017 100 kHz, (**c**) 2016 200 kHz, and (**d**) 2016 400 kHz datasets.

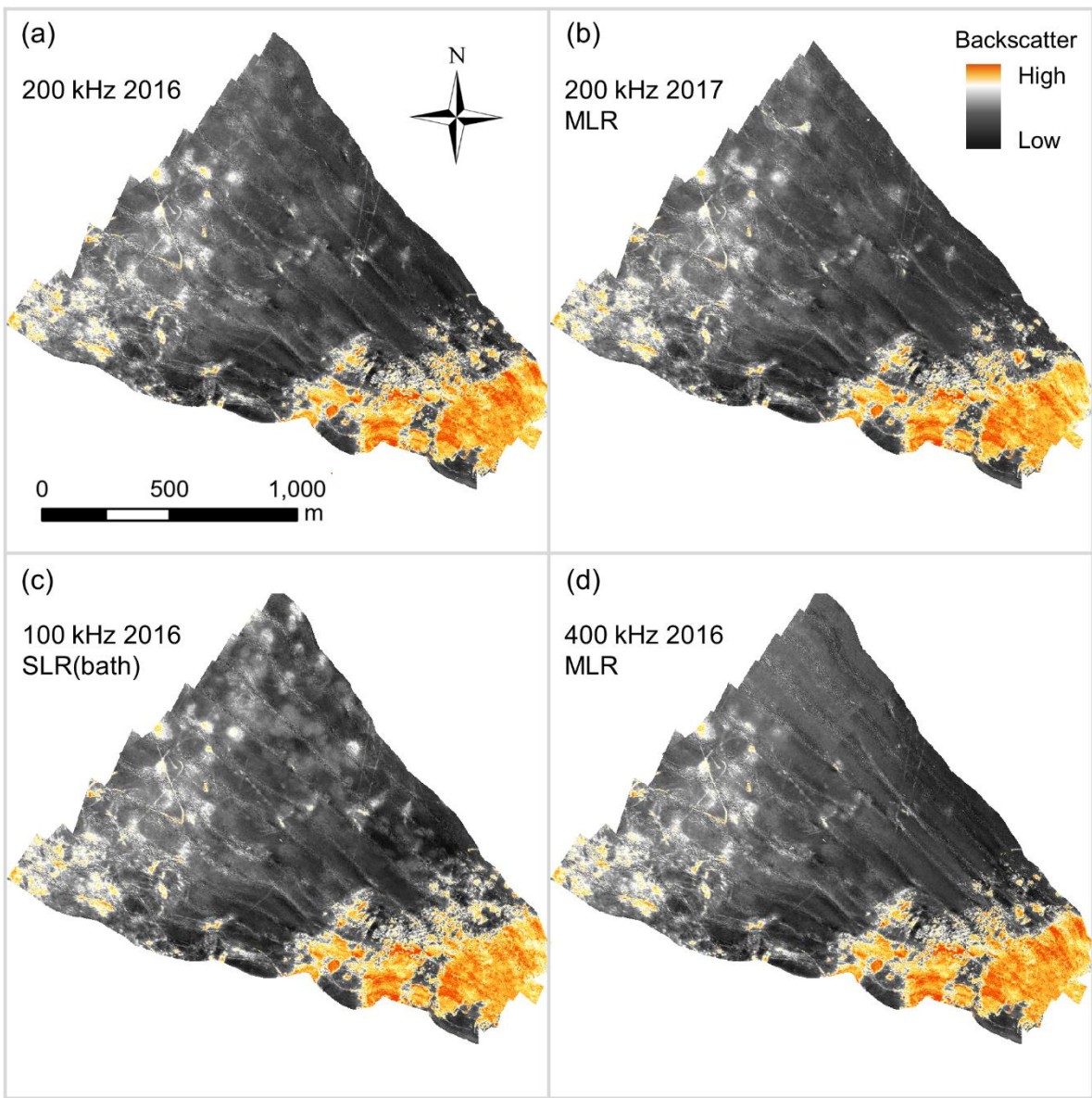

**Figure A3.** Highest quality bulk shift mosaics of (**a**) the 2016 Bedford Basin 200 kHz dataset, with (**b**) 2017 200 kHz, (**c**) 2016 100 kHz, and (**d**) 2016 400 kHz datasets.

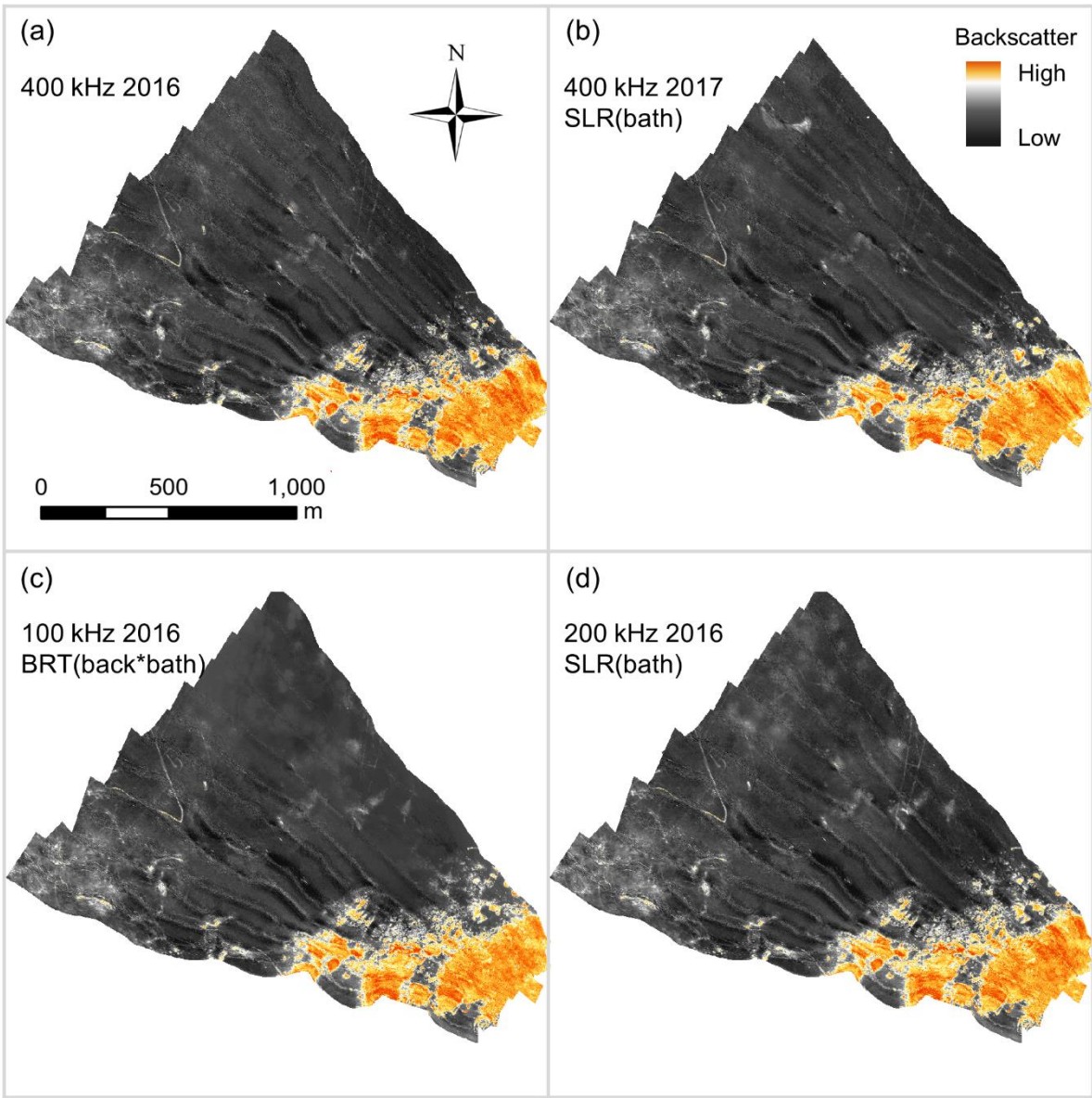

**Figure A4.** Highest quality bulk shift mosaics of (**a**) the 2016 Bedford Basin 400 kHz dataset, with (**b**) 2017 400 kHz, (**c**) 2016 100 kHz, and (**d**) 2016 200 kHz datasets.

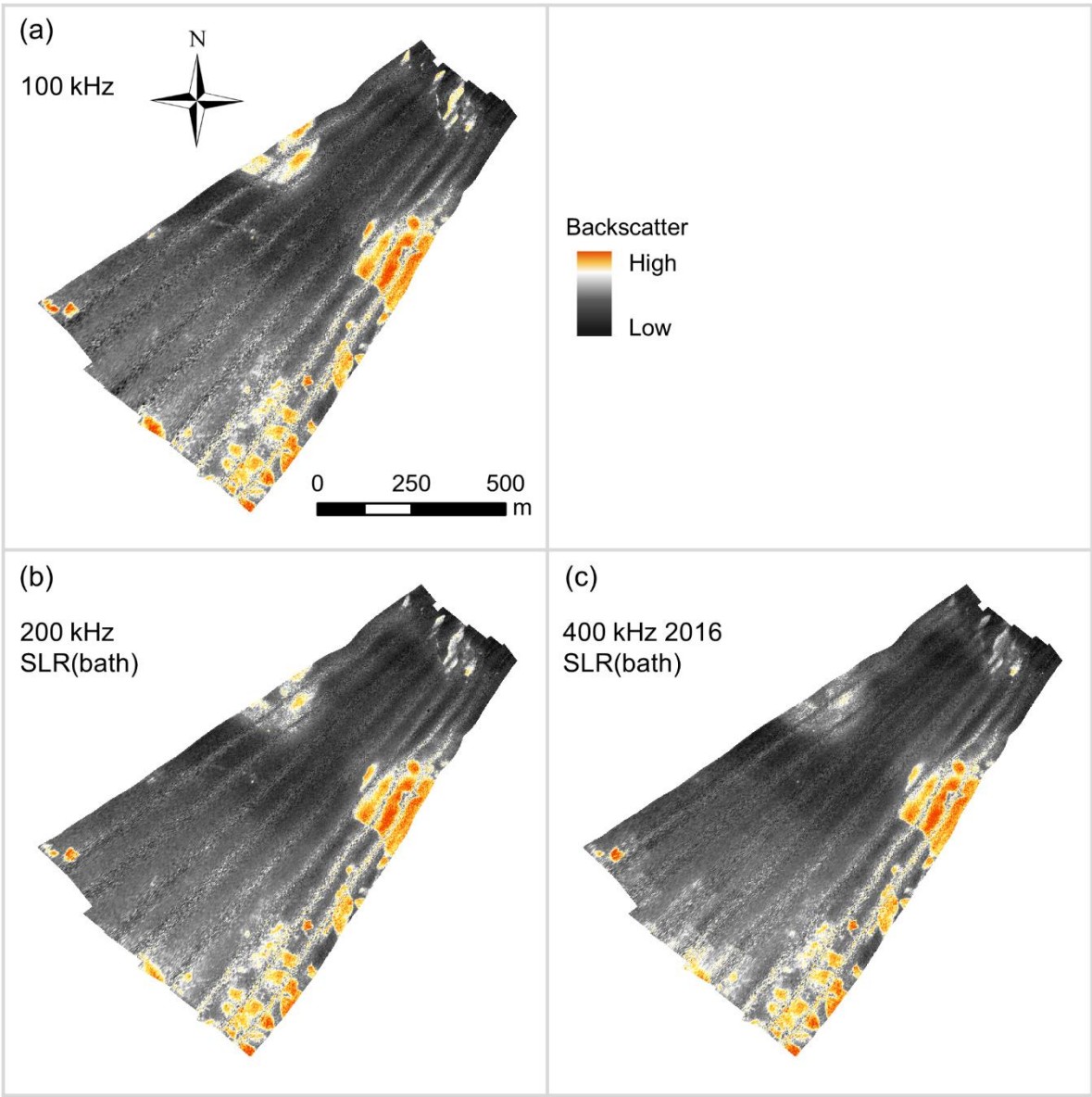

**Figure A5.** Highest quality bulk shift mosaics of (**a**) the Patricia Bay 100 kHz dataset, with (**b**) 200 kHz, and (**c**) 400 kHz datasets.

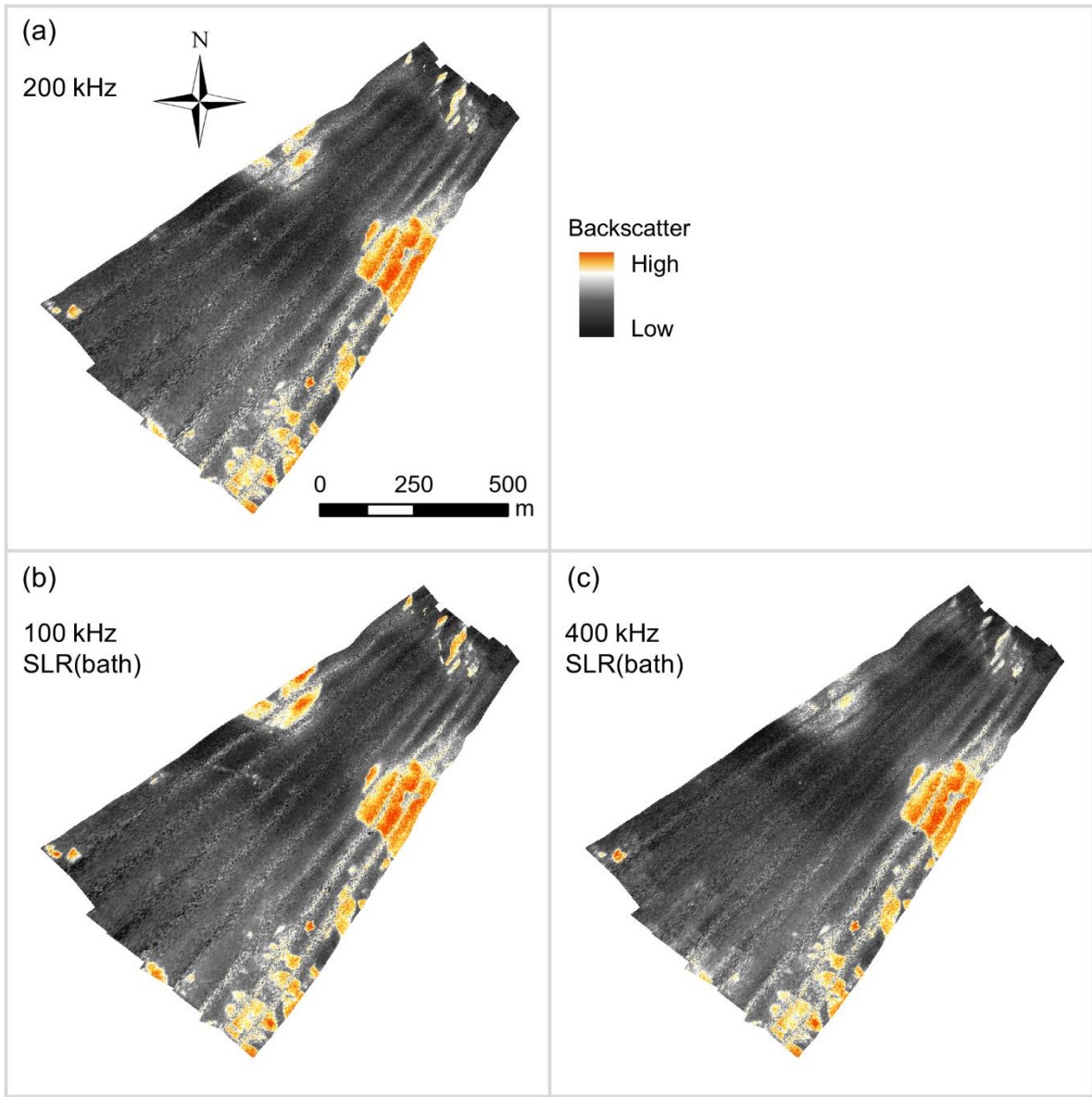

**Figure A6.** Highest quality bulk shift mosaics of (**a**) the Patricia Bay 200 kHz dataset, with (**b**) 100 kHz, and (**c**) 400 kHz datasets.

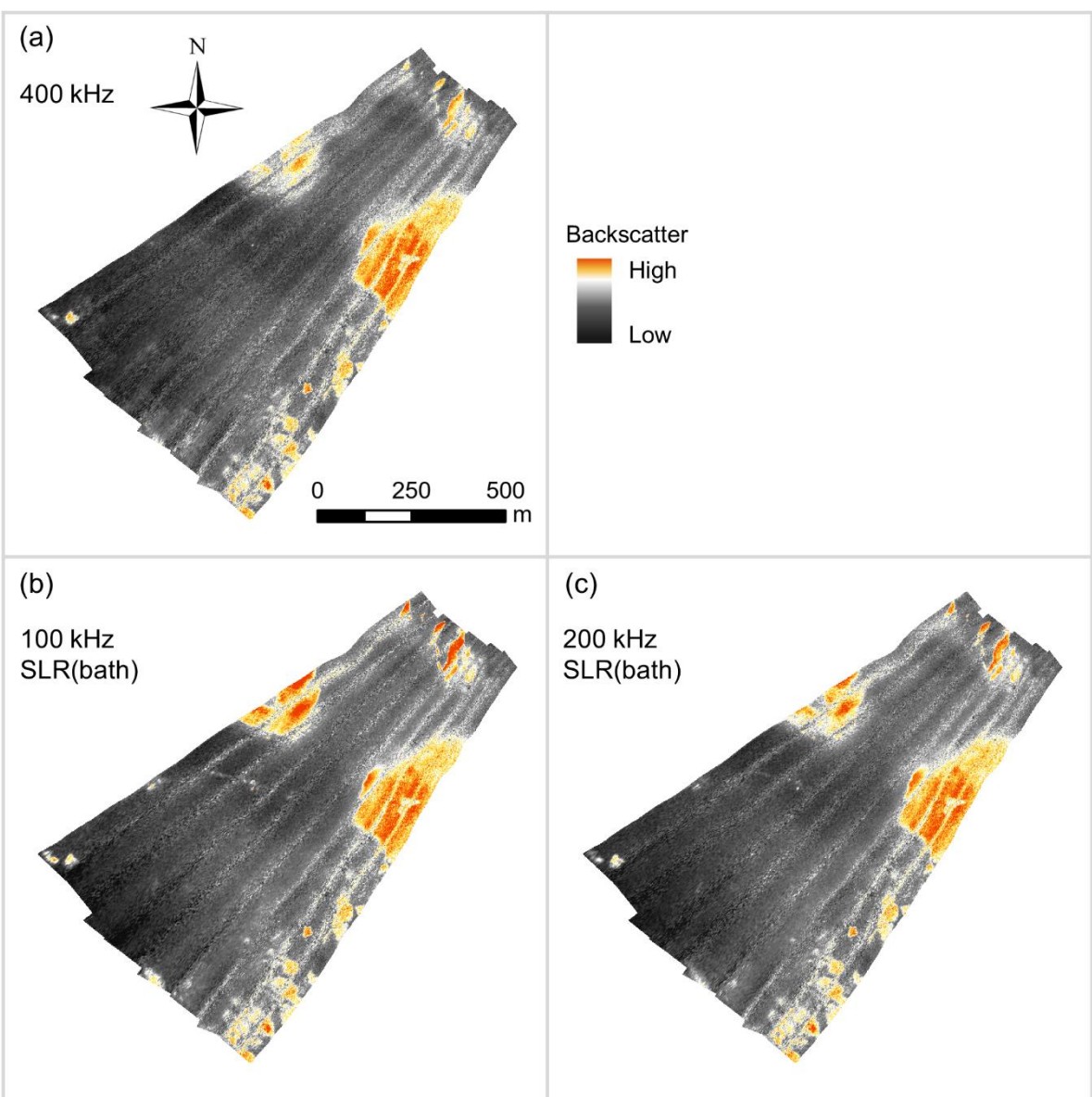

**Figure A7.** Highest quality bulk shift mosaics of (**a**) the Patricia Bay 400 kHz dataset, with (**b**) 100 kHz, and (**c**) 200 kHz datasets.

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
