# Peer review of "Harmonizing Multi-Source Sonar Backscatter Datasets for Seabed Mapping Using Bulk Shift Approaches"

_remotesensing, doi:10.3390/rs12040601_

Round 1

Reviewer 1 Report

Dear authors,

this is a very nice article presenting for first time a simple but efficient way to harmonize MBES backscatter datasets from multiple surveys involving the same or even different equipment. Sufficient examples are provided as test cases spanning Bs datasets of similar or different frequencies and the success of the harmonization is controlled over a variety of statistical derivatives. A python tool for the validation of the generated results is provided and this is considered a significant contribution to the acoustic habitat mapping community.

Although in some parts over-analytical, I could follow the manuscript without any question having been left open. The methods are clear and the conclusions are absolutely justified by the results. The discussion is covering all aspects that a researcher could expect about the methods involved.The language is descent and I personally couldn't find any spell issues.

The only recommendation that I would please you to consider regards the applicability of this exact methodology to Bs datasets from other sourcecs rather than MBES. Sidescansonars, interferometric echosounders or even synthetic apperture sonars are widely employed for habitat mapping, especially in shallow coastal waters. The described method seems to be appropriate for multisource Bs data from this kind of equipment and I strongly recommend putting some words in the introduction and the conclusions as a further, quite direct indeed (excluding maybe using bathymetry), application of your method. Refer to works from Tamset et.al., 2019 and Fakiris et.al., 2019 or others regarding the use of multi-frequency and/or multi-sonar surveys towards seafloor mapping. They show that sidescan sonar mosaics can undergo various levels of radiometric corrections, but in the luck of a perfect acoustic physical model, they can never be absolutely consistent between different surveys (as you also discussed about MBES). A method like yours could potentially be highly desired for multisource sidescan sonar datasets spanning various temporal and spatial scales and overlaps, for coverage maximization or even for change detection.  

Finally, I would recommend putting some regression lines on top of a scatter plot within your main text body, in order to let the reader understand the difference between the various regression models considered, including an example of over-fitting.

Great job!

Good luck!

Author Response

Thanks to the reviewer for taking the time to read our manuscript. Their comments are appreciated, and the following changes have been implemented based on these.

"The only recommendation that I would please you to consider regards the applicability of this exact methodology to Bs datasets from other sourcecs rather than MBES...  I strongly recommend putting some words in the introduction and the conclusions as a further, quite direct indeed (excluding maybe using bathymetry), application of your method... a method like yours could potentially be highly desired for multisource sidescan sonar datasets spanning various temporal and spatial scales and overlaps, for coverage maximization or even for change detection."

This is a great point. We have referenced the literature suggested by the reviewer on multi-source side-scan sonar in the introduction (line 105 in the revised manuscript), and have mentioned in the discussion that these methods are also applicable to backscatter datasets from other sensors (lines 467-469).

"Finally, I would recommend putting some regression lines on top of a scatter plot within your main text body, in order to let the reader understand the difference between the various regression models considered, including an example of over-fitting."

We feel that the number of figures in the manuscript is currently quite high, and we hesitate to add another if it is not absolutely necessary. Examples of the various regression models, and of statistical overfitting, have been provided in the supplementary within the R function tutorial. A specific overfitting example (dynamic range compression) has also been provided in Appendix B. We feel that this is an appropriate compromise to provide the interested reader with examples, but avoid cluttering the main text.

Reviewer 2 Report

The manuscript "Harmonizing multi-source sonar backscatter datasets for seabed mapping using bulk shift approaches" by Misiuk et al. describes different procedures to harmonise backscatter mosaics recorded by different systems, with different frequencies and at (with assumptions) different time. 

This is probably the shortest review I have written so far. The paper is very well written, easy to follow and thoroughly addresses an important topic that many of us in habitat mapping currently face.

It is also highly appreciated that the processing scripts are available as supplement and on GitHub (I did not test the code or the tutorial so far due to time constraints).

I don't know whether the R2Sonic dataset is openly available. If this is the case, maybe a link to the data, or even the processed mosaics could be added to the supplemental materials? This would make the paper fully reproducible, and would allow others in the future to extend on the work of this study, or compare the results. Probably there will be further studies, e.g. on the depth-backscatter relationship down the road. 

Congratulations to the authors. 

Author Response

We appreciate the reviewer's feedback regarding our manuscript. We agree that sharing the multispectral data is highly desirable to facilitate reproducibility. The raster files are too large to be deposited as supplementary material in the journal (>180 MB), but R2Sonic (the data provider) has indicated that we may share the data. We have changed the supplementary material text to indicate that the data can be obtained from either R2Sonic or the authors.

Reviewer 3 Report

The reviewed article concerning harmonizing multi-source sonar backscatter datasets for seabed mapping presents results of research on three multispectral MBES datasets used to simulate the harmonization of backscatter collected over multiple years using multiple operating frequencies.
The structure of the article is considered and clear, divided into three parts: the main part of the article, appendixes and supplementary files.
In the introduction, the background and comprehensive review of the problem's literature were presented. The authors presented the equipment of the hydrographic system and location of surveys. MBES data processing and mulating harmonization have been characterized. On the basis of the research, harmonizing different surveys (same frequency) and different frequencies have been processed. Results of the research have been presented in graphic form. Conclusions, on the basis of the research, are clear.

Author Response

We appreciate this feedback and thank the reviewer for taking the time to read our manuscript.